# Efficient direct repairing of lithium- and manganese-rich cathodes by concentrated solar radiation

Hailong Wang[1,2], Xin Geng[1], Linyu Hu[2], Jun Wang [3], Yunkai Xu[4], Yudong Zhu [5], Zhimeng Liu[1], Jun Lu [4] ✉, Yuanjing Lin [2] ✉ & Xin He [1,6,7] ✉

Lithium- and manganese-rich layered oxide cathode materials have attracted extensive interest because of their high energy density. However, the rapid capacity fading and serve voltage decay over cycling make the waste management and recycling of key components indispensable. Herein, we report a facile concentrated solar radiation strategy for the direct recycling of Lithium- and manganese-rich cathodes, which enables the recovery of capacity and effectively improves its electrochemical stability. The phase change from layered to spinel on the particle surface and metastable state structure of cycled material provides the precondition for photocatalytic reaction and thermal reconstruction during concentrated solar radiation processing. The inducement of partial inverse spinel phase is identified after concentrated solar radiation treatment, which strongly enhances the redox activity of transition metal cations and oxygen anion, and reversibility of lattice structure. This study sheds new light on the reparation of spent cathode materials and designing high-performance compositions to mitigate structural degradation.

The lithium (Li)- and manganese (Mn)-rich layered oxide materials (LMRO) are recognized as one of the most promising cathode materials for next-generation batteries due to their high-energy density[1]. Both transition metal cations(*TM*s) and oxygen (O) anions are involved in the redox reaction, which enables a specific capacity larger than 250 mAh g$^{-1}$ and an average discharge voltage of 3.6V[2]. However, the redox of lattice oxygen leads to the formation of localized defects, oxygen loss in the form of gas release from the surface, and migration of *TM*s[3–5], resulting in low initial Coulombic efficiency, rapid capacity fading, and voltage decay. Therefore, practical applications of LMRO

need to solve these drawbacks and meet waste management challenges at the end of the device's useful life.

There have been great efforts, including surface modification[6,7] and intrinsic structural design[8,9], to enhance electrochemical stability. For surface modification, gas-solid interfacial reaction (GSIR) is used to uniformly create oxygen vacancies on the surface without affecting the original structure, which improves the initial Coulombic efficiency and cycling stability of electrodes[10]. Besides, the spinel phase is introduced into the layered structure to improve Li-ion diffusion in three-dimensional channels, and the close-packed oxygen arrangement in

[1]School of Chemical Engineering, Sichuan University, Chengdu 610065, China. [2]School of Microelectronics, Southern University of Science and Technology, Shenzhen 518055, China. [3]School of Innovation and Entrepreneurship, Southern University of Science and Technology, Shenzhen 518055, China. [4]College of Chemical and Biological Engineering, Zhejiang University, Hangzhou 310027, China. [5]Department of Materials Science and Engineering, Shenzhen Key Laboratory of Full Spectral Solar Electricity Generation (FSSEG), Southern University of Science and Technology, No. 1088, Xueyuan Rd, Shenzhen, Guangdong 518055, China. [6]College of Electrical Engineering, Sichuan University, Chengdu 610065, China. [7]College of Civil Aviation Safety Engineering, Civil Aviation Flight University of China, Guanghan 618307, China. ✉e-mail: junzoelu@zju.edu.cn; linyj2020@sustech.edu.cn; xinhe@scu.edu.cn

the new phase suppresses oxygen release with alleviated transition metal migration and structural degradation[11].

All these strategies aim to push the envelope of electrochemical performance, but few studies have considered the post-handling process, such as repairing or regeneration of cycled LMRO. The simple heat treatment without re-lithiation will induce severe *TM* migration, and cause the layered structure to slowly relax/collapse to rock-salt phase because the cycled LMRO is a metastable structure. In contrast, concentrated solar radiation (CSR) with concentrated light can provide ultra-strong radiation and ultra-high temperature within a short time, which provides efficient energy to repair spent LMRO with limited side reactions. Compared to traditional thermal annealing, CSR possesses a combination of photocatalysis and photothermal benefits[12,13]. Given that the surface of LMRO-50C is mainly constructed by $LiMn_2O_4$ and $MnO_2$, which can absorb photos in the wavelength ranging from ultraviolet to infrared to generate hole-electron pairs[14–17]. Thus, they work as an induce layer for the generation of holes and are responsible for their transfer in the lattice. In addition, the oxygen vacancies formed in LMRO-50C as the result of electrochemical cycling can effectively mitigate the recombination of hole-electron pairs and facilitate the transfer of holes to the inside of the particle[18,19].

In this work, we directly and effectively repaired degraded LMRO materials using the CSR process. The discharge capacity of the repaired electrode successfully returns to the original level due to the reactivation of the redox activity of both *TM*s and O-anion. Beyond that, CSR significantly suppresses the fast capacity fading and voltage decay. The inverse spinel phase is efficiently introduced into the bulk of particles and forms a co-existence of layered-spinel structure. Such a unique structure can maintain phase stability and lower the internal strains, which assures a stable cycling performance. Our work provides a new strategy to recycle and repair spent LMRO material. Moreover, it also offers to access extra capacity after direct electrode reparation.

## Results

### Structural and morphological evolution of cycled LMRO after CSR treatment

The optical path of the solar light simulation device is shown in Fig. 1a, and the simulated solar light spectrum is shown in Supplementary Fig. 1. Light is concentrated by a special Fresnel lens matrix to offer concentrated light spots with a uniformly distributed matrix pattern in the working plane. The light intensity along the vertical direction of a single focused light beam is measured by a light intensity sensor to provide irradiance details. During the CSR process, the working plane is located at the horizontal plane which is 5 mm below the focused point (Fig. 1a), and the reaction spots experience strong irradiation and high temperature within a short period. During the period of the CSR process, the LMRO electrode is slowly moved into the horizontal working plane to experience direct light incident and scattered light radiation, as well as thermal transmission in the electrode, the center temperature of each focus point in the focus array is set as 350 °C to avoid decompositions of the binder and conductive carbon black. LMRO experiences obvious loss of Li-ion and slightly deficiency of *TM*s (Ni, Co) through the CSR treatment, from $Li_{1.014}Mn_{0.56}Ni_{0.15}Co_{0.13}O_2$ (LMRO-50C) to $Li_{0.762}Mn_{0.56}Ni_{0.13}Co_{0.11}O_2$ (LMRO-50CS), as quantified by inductively coupled plasma optical emission spectroscopy (ICP-OES) in Supplementary Table 1. The SEM images (Fig. 1c, d and Supplementary Fig. 2) demonstrate the morphological changes of electrode before and after the CSR treatment. After 50 cycles, LMRO particles are covered by a dense cathode–electrolyte interface (CEI) layer composed of electrochemical decomposition products (Fig. 1c). The CEI layer was effectively partially removed by the CSR treatment and the morphology of the electrode was not disrupted.

X-ray powder diffraction (XRD) is utilized to determine the crystal structure evolution of LMRO after cycling and CSR treatment. The

differences in crystal structure between the pristine LMRO (LMRO), LMRO after 50 cycles (LMRO-50C), and repaired LMRO (LMRO-50CS) are marked in XRD patterns in Fig. 1e and the refined patterns of three samples are shown in Supplementary Fig. 3-5. The (003), (101) and (104) peaks shift to the lower angle and all the peaks are broadened after cycling. The shift of diffraction peaks is attributed to the lattice expansion which is caused by irreversible Li ion loss and formation of new phases (such as spinel phase and rock salts phase) after cycling[20]. The broadening of peaks is related to the increase of *TM*s disorder in local lattice and the decrease of domain size. With the destruction of the honeycomb $LiMn_6$[21], the characteristic peaks in the 20°-25° are completely disappeared over cycling. Although previous studies have proved that the original superlattice structure can be recovered from the high-temperature re-lithiation of cycled LMRO materials[4,22,23], no superlattice peaks are observed from the XRD pattern of the LMRO-50CS sample. However, the refinement results (Supplementary Fig. 5) and the appearance of the (220) characteristic diffraction peak (2Theta = 30.5°) indicate that the CSR process induced the production of a large number of spinel structures accompanied by partial inverse spinel[24]. Due to the partial occupation of Li atoms at 8a sites and Mn atoms at 16d sites in both spinel $LiMn_2O_4$ and $Li_4Mn_5O_{12}$, the (220) diffraction peak around 31° commonly appears weak or masked[25]. In LMRO-50CS sample, the enhancement of the (220) peak is attributed to the rearrangement of local atoms in inverse spinel structure, which is akin to $LiNiVO_4$, where V atoms occupy the 8a sites in the crystal structure and Li/Ni atoms equally share the 16b site[26]. In the case of LMRO-50CS, higher peak intensity corresponds to more *TM* atoms migrating the 8a sites and more Li atoms occupying the 16d sites. The Raman spectra (Supplementary Fig. 6) also confirm this fact, where the characteristic Raman peak at 658 $cm^{-1}$ attributed to the spinel phase[27] on surface is significantly enhanced after CSR treatment. As a result, locally disordered spinel structures (inverse spinel phase) are induced into the layered host, which can suppress the Jahn-Teller effect and improve the structural stability[28]. Based on these characterization results, the schematic diagram shown in Fig. 1f demonstrates the compositional variation of LMRO during the electrochemical process[29] and after the CSR treatment.

The HRTEM images of the LMRO-50C and LMRO-50CS in Fig. 1g and h demonstrate the evolution of local structure in detail. After 50 cycles, spinel and rock-salt phases are formed on the particle surface (Fig. 1g) due to irreversible oxygen release and surface reconstruction[22]. The combination of photocatalysis and photothermal effect during CSR treatment leads to large spinel domains on the surface and in the bulk, which causes the coexistence of layered-spinel phases throughout the particle (Fig. 1h). Despite the bulk of the particle is still mainly constructed by layered structure, the existence of spinel grain in domain 1 and 2, and the existence of mixed-phase prove substantial phase transformation during the CSR process.

### Surface components and elements valence changes of cycled LMRO after CSR treatment

The chemical components and elemental valence states of the particle surfaces undergo significant evolution during the CSR process. A variety of surface-sensitive detection techniques are performed to explore the evolution of particle surfaces in detail. To elucidate the mechanism of reactions that occurred on CEI layer during the CSR process, X-ray photoelectron spectroscopy (XPS) spectra are collected on LMRO, LMRO-50C, and LMRO-50CS electrode, as shown in Fig. 2a–d. All three samples show a strong C-C peak located at 284.8 eV, which is attributed to the conductive carbon black (Super-P) in electrode. LMRO-50C sample shows stronger C-O and C=O peaks at 286.6 eV and 288.8 eV in C 1*s* spectra, and 533.6 eV and 532.6 eV in O 1*s* when compared with LMRO, which attribute to organic components like $ROCO_2$-Li, R-$CF_x$, and R-$CO_xF_y$, polycarbonates in CEI layer formed after cycling[24,30,31]. In the F 1*s* spectra, a relatively high

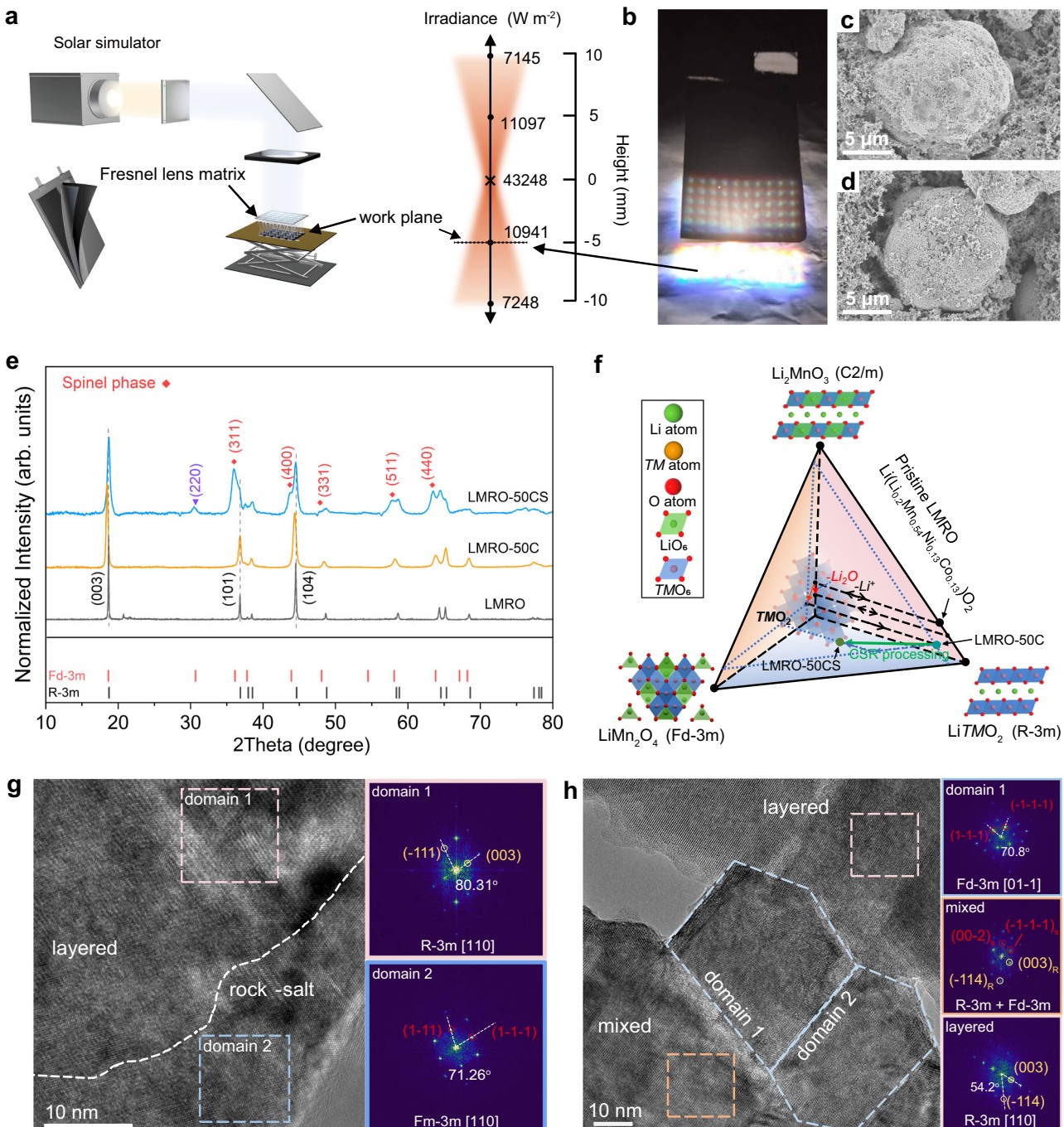

**Fig. 1 | Direct reparation of electrode via CSR process. a** Schematic diagram for the optical path of the solar light simulation device and the light intensity distribution at the focus point, the height of the working plane (dotted line) is 6 mm below the focus point. **b** Real image of LMRO electrode under CSR treatment through Fresnel lens matrix concentrated solar light. SEM images of LMRO-50C (**c**) and LMRO-50CS (**d**). **e** XRD patterns of LMRO, LMRO-50C and LMRO-50CS, the red markings correspond to the spinel phase and the purple markings represent the inverse spinel phase produced after CSR treatment. **f** Schematic diagram for the compositional changes of corresponding phases after the electrochemical cycles[29] and CSR treatment. HRTEM images of LMRO-50C (**g**) and LMRO-50CS (**h**) and the corresponding fast Fourier transform (FFT) patterns. The subscript character "S" represents the spinel phase and "R" represents the layered phase.

proportion of inorganic components, including $Li_xPF_y$ (686.8 eV), $Li_xPO_yF_z$ (685.6 eV), and LiF (684.7 eV), are observed on the LMRO-50C surface, which attribute to the decomposition of $LiPF_6$ salt[24]. The enrichment of LiF (55.8 eV) and $Li_2CO_3$ (55.2 eV) in CEI appears in the Li 1$s$ spectrum as well, which agrees with findings in the previous report[32]. In comparison with LMRO-50C, the C-O and C=O peaks observed in the C 1$s$ and O 1$s$ spectra nearly vanished in the LMRO-50CS sample. This disappearance is attributed to the decomposition of organic components within the CEI, induced by localized high

temperatures during the CSR process. It is noteworthy that the peaks representing the inorganic components in the CEI layer do not change significantly due to the better thermal stability of these inorganic components[24,33]. As a result, the CEI layer of LMRO-50CS sample becomes rich in inorganic components like LiF, $Li_2CO_3$, $Li_xPF_y$, and $Li_xPO_yF_z$, which contribute to the improved cycling stability of the repaired material[34,35].

The oxidation states of *TM*s cations and O anion reveal the reconstructed species on electrode surface at different states. As

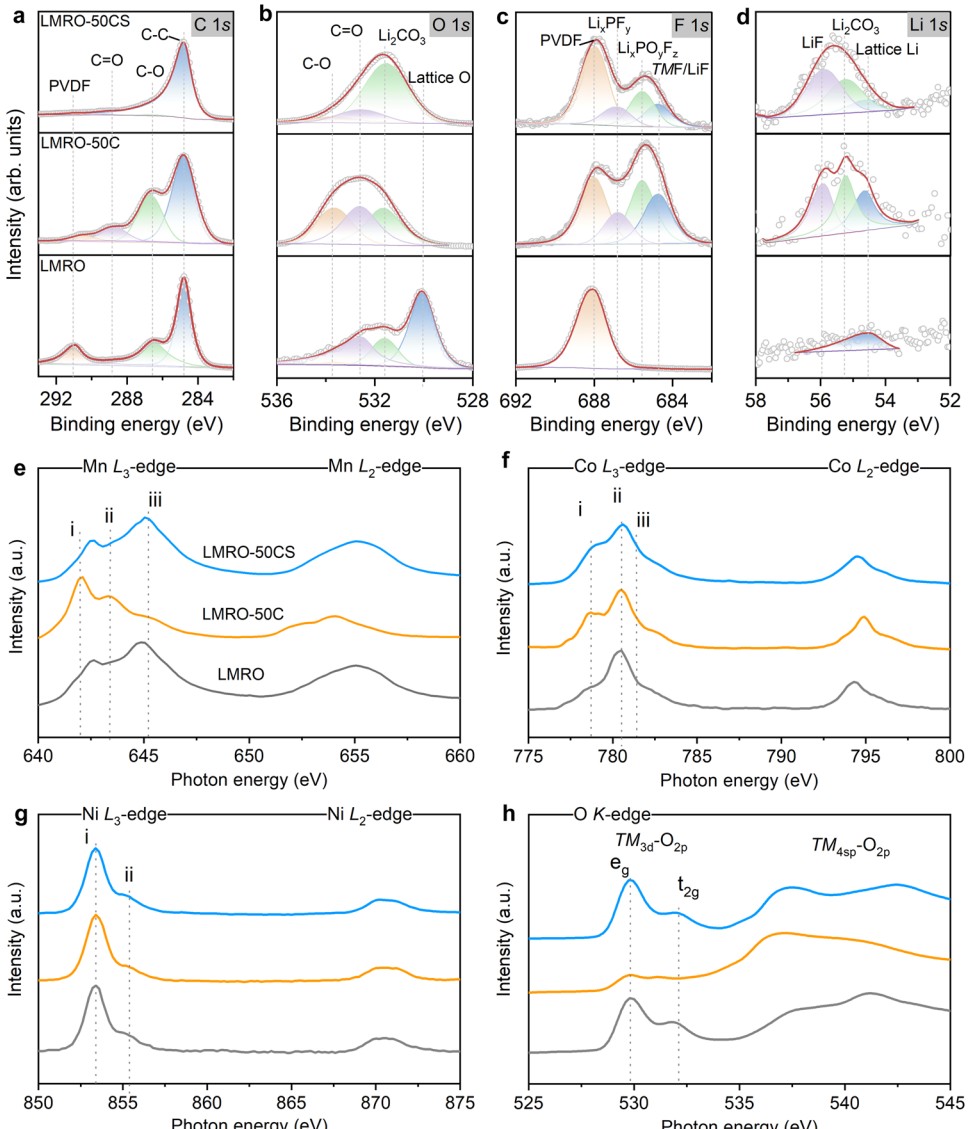

**Fig. 2 | Surface composition evolution from LMRO to LMRO-50C to LMRO-50CS.** XPS spectra of C 1s (**a**), O 1s (**b**), F 1s (**c**), and Li 1s (**d**), for LMRO, LMRO-50C and LMRO-50CS samples. sXAS patterns of Mn L-edge (**e**), Co L-edge (**f**), Ni L-edge (**g**) and O K-edge (**h**) collected in total electron yield (TEY) mode.

shown in Fig. 2e–h, soft X-ray absorption spectroscopy (sXAS) spectra are collected from LMRO, LMRO-50C, and LMRO-50CS samples in TEY mode. The appearance of low energy shoulder (peak i) for both Mn $L_3$-edge and Co $L_3$-edge is significantly increased after 50 cycles, indicating that $TM_{3d}$ bonds to higher occupancies and the emergence of $Mn^{2+}$ and $Co^{2+}$ upon cycling[36]. With further CSR treatment of the cycled electrode, the peak i in the Mn $L_3$-edge of LMRO-50CS is greatly reduced while the peak iii is significantly enhanced and shifted slightly to a higher energy region, showing a similar line shape as $MnO_2$. This suggests that Mn is oxidized to tetravalent accompanied by the extraction of Li-ion during the CSR treatment. For the Co $L_3$-edge, the decrease of peak i associated with a high-energy shift of peak ii indicates the oxidation of Co. Different from the oxidation of Mn and Co, Ni maintains as divalent even after the CSR process due to the fact that the oxidation potential of Ni locates outside of the LMRO-50C[1], leading to the same valence states with the pristine state and LMRO-50C sample.

Generally, the intensity and line shape of the pre-edge spectrum of O K-edge are dominated by the density of unoccupied $TM_{3d}$-$O_{2p}$ hybridized states, and the broad bond corresponds to the hybridization between $TM_{4sp}$ and $O_{2p}$, respectively[37]. The O K-edge spectra of

LMRO shown in Fig. 2f is very similar to that of $MnO_2$, featuring a strong pre-edge peak, which is attributed to the high concentration of manganese and the great number of unoccupied 3d orbitals in $Mn^{4+}$. After 50 cycles, the peak intensity variation can be explained from two aspects: Firstly, the reduction of Mn/Ni and the formation of spinel/rock-salt phases on the surface during cycling[36]. Secondly, the formation of oxygenated by-products (such as $Li_2O$, $Li_2CO_3$, LiOH, RO-Li and $ROCO_2$-Li) from the electrolyte decomposition during the cycling[38]. No available orbitals exist to hybridize with $O_{2p}$ orbital in these oxygen-containing byproducts, thus resulting in a weak pre-edge peak feature[39]. Furthermore, the strong hump at around 536 eV stands for the existence of organic species as parts of the CEI component on the particle surface[40]. Notable, the pre-edge pattern after CSR treatment closely relates to the hybridization of TM-O and the decomposition of surface by-products. During the CSR process, the spatial Li-ion loss evenly occurred through the host material particle, which has been confirmed by the TOF-SIMS mapping of Li (shown in Supplementary Fig. 7) and its corresponding radial distribution function (inset graphics in Supplementary Fig. 7). Simultaneously, electrons are removed from 3d orbitals of Mn and Co, and lead to their oxidization. Oxidation of Mn and Co provides the driving force for electrons to transfer from

$O_{2p}$ to *TM* band to compensate for the charge imbalance, which leads to the increase of pre-edge peak[41,42]. The decrease in the peak intensity at 536 eV is attributed to the decomposition of the by-products during CSR treatment, which is consistent with the results of the previous SEM images (Fig. 1d) and XPS spectra (Fig. 2a-d). In addition, the decomposition of by-products and organic components in CEI layer is also supported by the thermogravimetric (TG) analysis. The TG and DTG profiles of the electrodes are displayed in Supplementary Fig. 8. All samples had two wide DTG peak groups at 350 and 510 °C, representing two different stages of weight loss. The first weight loss stage from 300 to 420 °C can be assigned to a complex residual carbonate on the surface at 330 °C[43] and part of PVDF binder decomposition occurs at 350 °C[44,45]. The second weight-loss stage from 460 to 560 °C belongs to the decomposition of conductive carbon black (Super-P)[46]. For the LMRO-50C, a weight loss peak appeared early at 340 °C, which can be assigned to the decomposition of partial CEI components. Although the working temperature of the CSR process already reaches

close to the decomposing temperature of PVDF, the decomposition of the PVDF binder is limited due to the rapid treatment. This is also confirmed by the XPS results, where the LMRO-50CS sample still shows a strong representative peak of PVDF at 688 eV in F 1*s* spectra (Fig. 2c).

## Electrochemical performance and structural evolution

The electrochemical performance of LMRO and its recycled electrode are shown in Fig. 3. The capacity of conventional LMRO reduces down to 220 mAh g⁻¹ after 50 cycles at C/2 (1 C = 250 mAh g⁻¹), and continues fading during cycling. When CSR treatment is directly utilized on the cycled electrode, the delivered discharge capacity recovers to 273 mAh g⁻¹ and the fading trend is effectively mitigated. However, an unusual initial Columbic efficiency could also be observed from the LMRO-50CS sample, corresponding to the lower Li content in the repaired $Li_{0.762}Mn_{0.56}Ni_{0.13}Co_{0.11}O_2$ (Supplementary Table 1). Since there is a large number of Li vacancies in the crystal structure, more Li-

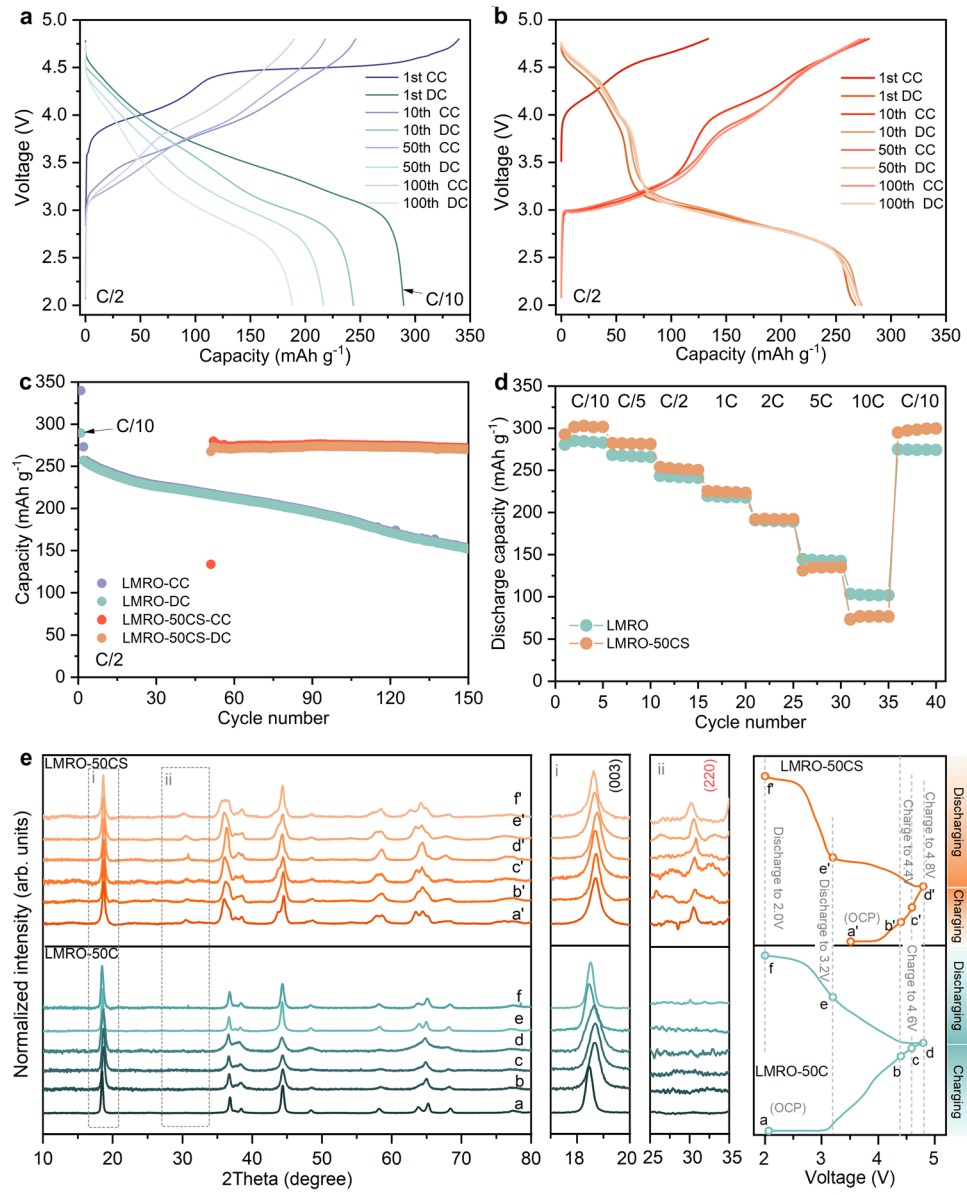

**Fig. 3 | Electrochemical performance and Ex-situ sXRD measurement.** The charge-discharge profiles of LMRO-50C (**a**) and LMRO-50CS (**b**) electrode. The cycling performance at C/2 (**c**) and rate capability (**d**) for LMRO and LMRO-50CS. **e** Ex-situ sXRD of LMRO-50C and LMRO-50CS samples at representative voltage with the enlarged view in the range of 2Theta = 17.5-20° and 25-35°, and the corresponding selected voltage stages in the first cycle profiles. The (CC) in **a**–**c** represents the Charging state and the (DC) represents the Discharging state. 1 C = 250 mAh g⁻¹.

ion can be reinserted during the subsequent discharge. For the second cycle, the charging capacity reaches up to 273 mAh g⁻¹ with the emergence of a long redox plateau in the voltage range between 3.0-3.3 V and 2.9-2.6 V, respectively. Such low potential plateau corresponds to the oxidation peak at 3.0 V and the reduction peak at 2.8 V in Supplementary Fig. 9b, which are often associated with the introduction of spinel phase into LMRO materials[11,47]. The LMRO-50CS has negligible voltage drop (Supplementary Fig. 9c) during cycling, and with high-capacity retention after 100 cycles (Fig. 3c, and Supplementary Fig. 9e). Since the rapid capacity decay and voltage decay are mitigated, LMRO-50CS can maintain a higher energy density than the LMRO for the long-cycling (Supplementary Fig. 9). A comparison of the rate performance of LMRO and LMRO-50CS is shown in Fig. 3d and Supplementary Fig. 10a, b. The LMRO-50CS delivered a higher discharge capacity until 1 C, but slightly lower at high rates. Electrochemical impedance spectroscopy (EIS) is an effective technique for probing the diffusion kinetics of Li-ion. The detailed calculations are shown in Supplementary Note 2. The Li-ion diffusion coefficients ($D_{Li}^+$) estimated from EIS spectra (Supplementary Fig. 10c, d) for LMRO and LMRO-50CS are $6.868 \times 10^{-17}$ cm² S⁻¹ and $1.936 \times 10^{-17}$ cm² S⁻¹, respectively. This suggests that the Li-ion diffusion of the LMRO-50CS is slightly restricted due to the long diffusion pathway in the Layered-spinel structure, thus leading to the deterioration of the rate performance.

The ex-situ synchrotron XRD (sXRD) spectra obtained during the first charge-discharge process are shown in Fig. 3e and the ex-situ XRD spectra obtained during the second charge-discharge process are shown in Supplementary Fig. 11a. The shifts of (003) peak is associated with the contraction and expansion of lattice $c$, and is caused by the delithiation and lithiation process. The lattice parameters acquired through Rietveld refinement at various voltages during the second charge-discharge process are shown in Supplementary Fig. 11b and detailed in Supplementary Table 2. Due to the presence of layered-spinel coherent structure, the LMRO-50CS samples significantly suppress the (003) displacement, which effectively suppresses the structural distortion and maintains the stability of the crystal lattice, thus enhancing the capacity stability and voltage stability during cycling. In addition, weak but highly reversible changes of (220) peak stand for the strong activity of the inverse spinel phase. Therefore, the electrochemical process proceeds under the mixture of multi-phase redox reactions.

## Evolution of redox couples during the electrochemical process and CSR process

To further investigate the reaction details with quantitative characterization on the charge compensation, the $L$-edge of *TM*s in electrodes are measured by sXAS in the total electron yield (TEY) and the total fluorescence yield (TFY) modes (shown in Fig. 4a–c). The

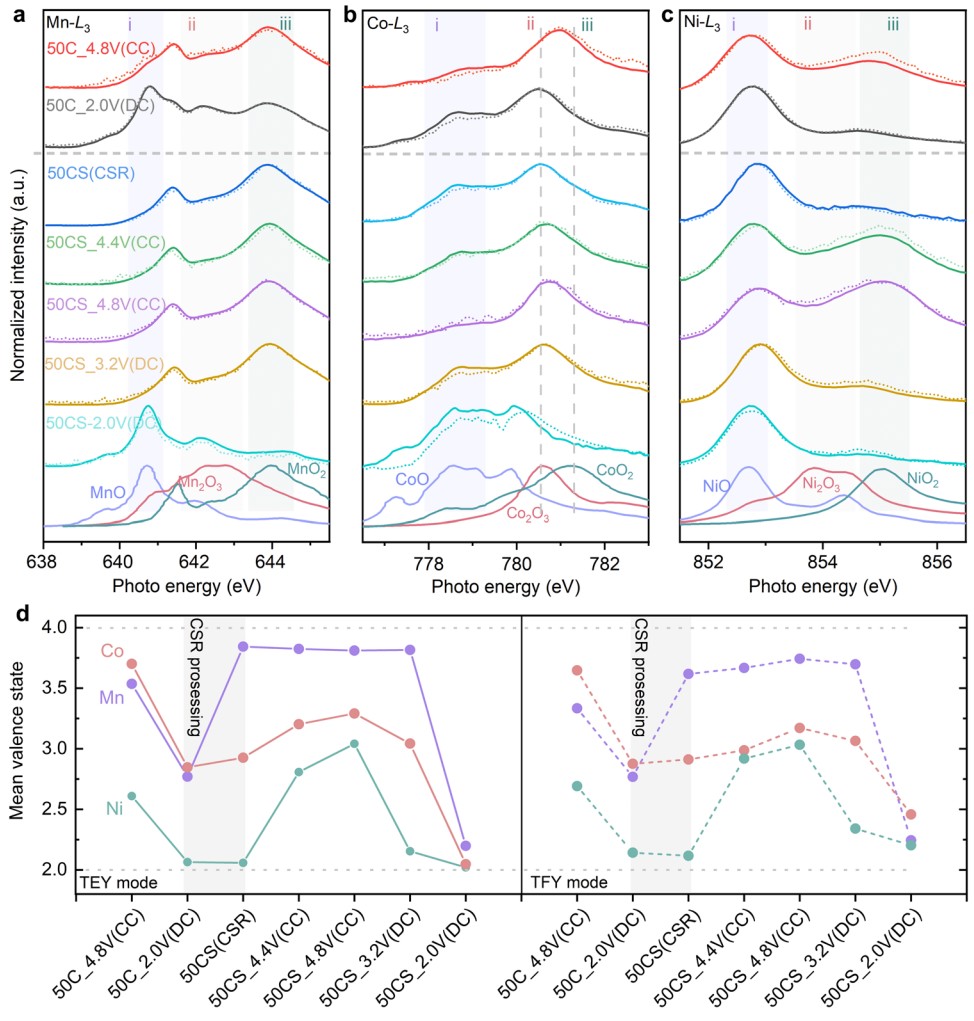

**Fig. 4 | Oxidation states quantification of *TM*s. a–c** The *TM*s (Mn, Co, Ni) -$L_3$ edge TEY (solid line) and TFY (dash line) spectra of electrodes at representative states. XAS spectra were normalized by the height between the highest peak and the background line. **d** The quantified mean oxidation states of each *TM*, based on the fitting results of the valence distributions in the spectra (**a–c**). The (CC) represents the Charging state, the (DC) represents the Discharging state and the (CSR) represents the sample after CSR treatment.

significant pattern variation indicates a severe evolution of redox couples from LMRO-50C to LMRO-50CS. Combining the spectral changes and fitting results, the quantified redox couples are $Mn^{3.33+}$/$Mn^{2.77+}$, $Co^{3.64+}$/$Co^{2.88+}$ and $Ni^{2.69+}$/$Ni^{2.14+}$ in the 50th cycle, which are summarized in Fig. 4d. The detailed analysis procedure is provided in Supplementary Note 1. With electrochemical cycling, the activity of Ni is reduced, while the contribution of Mn is substantially enhanced. Lower capacity contribution from redox of Ni can be attributed to the migration of Ni-ion from the bulk to the surface and dissolution in the electrolyte[36,48]. Furthermore, the reduction of Ni leads to the activation of more Mn and Co, resulting in a gradual voltage drop during cycling.

Since the Mn has been oxidized to 4+ after CSR treatment, no further oxidation of Mn can be identified from the LMRO-50CS samples as the voltage goes up to 4.8 V. Notably, neither Co nor Ni is oxidized to tetravalent, which is different from the oxidation capability of layered LMRO at initial cycles. During the discharge process, Ni is reduced to 2+, but Co and Mn are reduced to a much lower level than pristine. Peaks ii and iii disappeared in Mn $L_3$-edge when discharged to 2.0 V, and the shape of the spectrum is highly consistent with the reference spectrum of $Mn^{2+}$. This suggests that almost all the $Mn^{4+}$ has been reduced to $Mn^{2+}$. The characteristic peak of Co shifts to the lower energy range with spectra shape variation during the discharge

process, especially in the low voltage range from 3.2 V to 2.0 V, where more $Co^{3+}$ is reduced to $Co^{2+}$ at the surface. It can be concluded that the $Ni^{3+}$/$Ni^{4+}$, $Ni^{2+}$/$Ni^{3+}$and $Co^{3+}$/$Co^{4+}$ redox couples are mainly responsible for the reactions in the high voltage slope (4.8 V–4.0 V), whereas $Co^{2+}$/$Co^{3+}$ and $Mn^{2+}$/$Mn^{4+}$ redox couples are involved in the long plateau from 3.2 V to 2.5 V for the discharge of LMRO-50CS.

Besides the critical role of TMs, the contribution of O-anionic in LMRO-50CS also presents high relevance in the charge compensation. The high-angle annular dark-field scanning transmission electron microscope (HAADF-STEM) images for the cross-section of particles before and after CSR treatment and the corresponding EELS are shown in Figs. 5a and b. An obvious pre-peak feature transition can be observed through the particle of LMRO-50C material, the intensity of pre-peak gradually weakens from bulk to the surface, which finally disappears at the surface with a 20 nm gradient depth. However, the weak O pre-peak region is further expanded and an interval appears in the bulk for the LMRO-50CS particle. The weakening of the O pre-peak is closely related to the formation of O vacancies[49]. Thus, it can be concluded that a large number of O vacancies are formed as a result of the CSR process. The introduction of oxygen vacancies will decrease the DOS of the $O_{2P}$ band and reduce the covalency of the TM-O bond, resulting in enhanced reversibility of lattice oxygen redox during charge compensation[50,51].

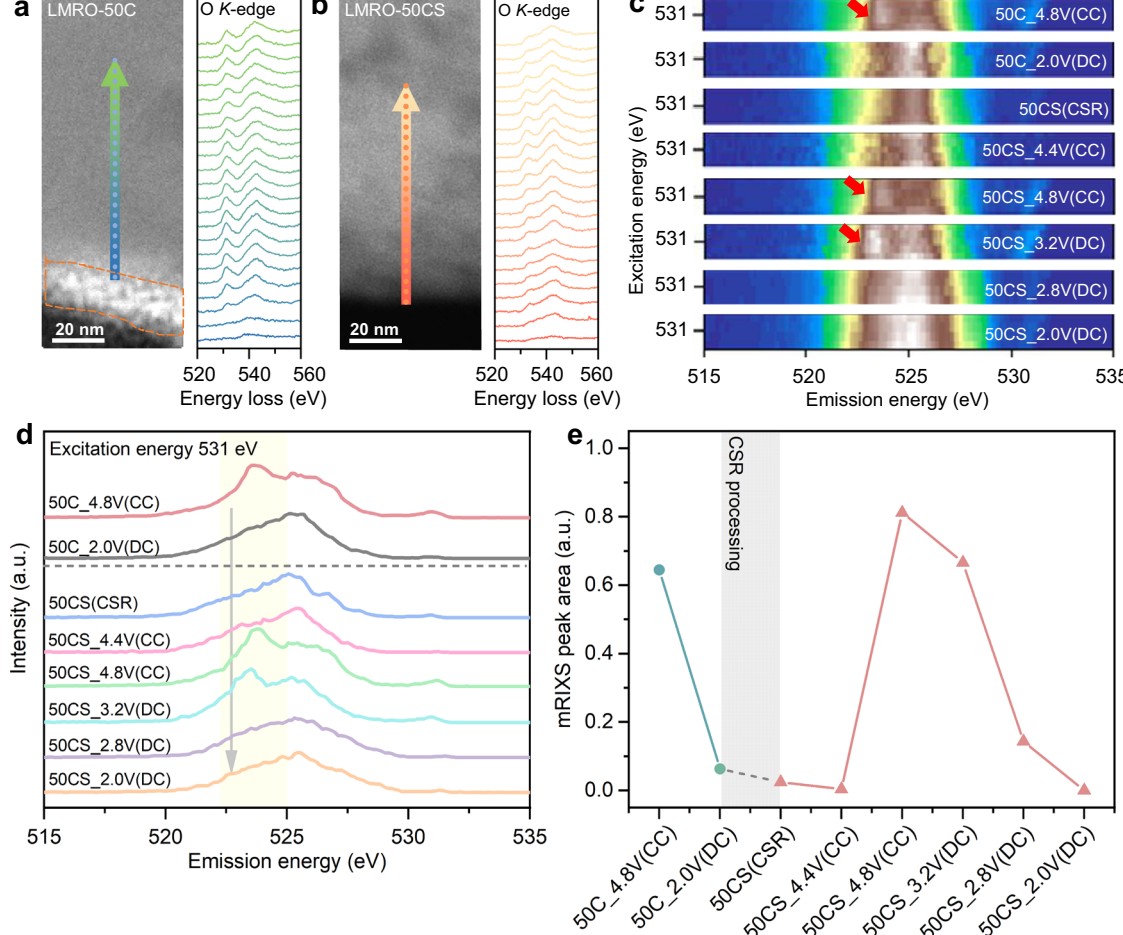

**Fig. 5 | Statistical and detailed characterization of O-anion.** HAADF-STEM and EELS O K-edge line scan spectra of LMRO-50C (**a**) (the area inside the dotted line is CEI species) and LMRO-50CS (**b**) along the direction of the arrow, from the surface to the bulk. **c** Local mRIXS map plotted at around 531 eV of LMRO-50CS and LMRO-50CS under different charge/discharge voltages, the red arrow marks the characteristics of oxygen oxidation that emerges at around 531 eV excitation energy and 523.7 eV emission energy. **d** Integrated RIXS intensity in characteristic energy range, from 530.5 to 531.5 eV. **e** The quantification of the integral area of the yellow shadow area in (**d**). The (CC) represents the Charging state, the (DC) represents the Discharging state and the (CSR) represents the sample after CSR treatment.

To quantify the oxygen performance, resonant inelastic X-ray scattering (mRIXS) technique is utilized to distinguish the intrinsic oxygen redox feature from the strong *TM*-O hybridization features within the same excitation energy range[37]. The mRIXS maps of selected region within the excitation energy of 530.5-531.5 eV are shown in Fig. 5c (full extraction energy maps are shown in Supplementary Fig. 12). When the electrode is charged to 4.8 V, the vertical and broad hybridization feature at around 525 eV emission energy is further broadened with the emerge of a feature at around 531 eV excitation and 523.7 eV emission energy, representing the emergence of oxidized oxygen (red arrow). The signatures of oxidized lattice oxygen can be identified from the sample of LMRO-50C charged to 4.8 V, LMRO-50CS charged to 4.8 V and discharged to 3.2 V. For a detailed analysis, the integrated mRIXS intensity is shown in Fig. 5d. The peak at 523-525 eV emission range can directly and quantitively represent the lattice oxygen oxidation. As is shown in both the mRIXS data of Fig. 5c and the integrated data of Fig. 5d, the characteristic signal of lattice oxygen oxidation appears when the LMRO-50C sample charging to 4.8 V and disappears discharging to 2.0 V. It is shown that the lattice oxygen is still involved in charge compensation after 50 cycles, although its capacity contribution continuously decreases. In the case of LMRO-50CS during the charging process, the signature of oxygen oxidation starts to emerge after charging to 4.4 V and shows the strongest peak at 4.8 V. This feature still exists when discharged to 3.2 V, and it is not annihilated until discharged to 2.8 V. The quantitative analysis of the yellow shadow area (Fig. 5d) is shown in Fig. 5e. The peak area variation extent is expanded after CSR treatment. This quantitative result provides strong evidence to support the hypothesis that CSR treatment enhances oxygen oxidation-reduction activity with high reversibility. The differential electrochemical mass spectroscopy (DEMS) (Supplementary Fig. 13) for the LMRO-50CS is consistent with the result of the mRIXS, with negligible oxygen release during the first charge at C/5. This provides strong evidence for the high reversibility of the lattice oxygen redox reaction.

Based on these results, we could conclude that the recovery of capacity mainly comes from the reactivation of redox from both *TM*s and O-anion. Since the change of valence states is closely related to the capacity contribution during charging and discharging[52], the increased valence state variation of *TM*s is the main reason for the capacity recovery, with Mn, in particular, constituting the majority of TMs in LMRO. It is puzzling that a large amount of reversible $Mn^{2+}$ in the full discharge state (Fig. 4a, d) when considering the fact that manganese participates in charge compensation mainly in the form of $Mn^{3+}/Mn^{4+}$ redox couple in LMRO materials[39]. The $Mn^{2+}/Mn^{4+}$ redox behavior is first observed in amorphous $Li_{1.5}Na_{0.5}MnO_{2.85}I_{0.12}$[53], and the reversible redox couple of $Mn^{2+}/Mn^{4+}$ is realized by high-valent cation ($Nb^{5+}$, $Ti^{4+}$) doping and partial $O^{2-}$ replacement by $F^-$ to form the disordered rock salt structure[54]. The existence of the $Mn^{2+}/Mn^{4+}$ redox couple not only contributes more capacity but also improves the stability of the structure. The 3d orbitals electronic configurations of $Mn^{2+}$ and $Mn^{4+}$ are $t_{2g}^3e_g^2$ and $t_{2g}^3e_g^0$, which possess a symmetrical electronic occupation with low-spin states. Thus suppresses the unfavorable Jahn-Teller distortion and stabilizes crystal lattices. Furthermore, the capacity contribution from the Co and Ni redox couple is transformed from $Co^{3+}/Co^{4+}$ and $Ni^{2+}/Ni^{4+}$ to $Co^{2+}/Co^{3+}$ and $Ni^{2+}/Ni^{3+}$ due to the abundant oxygen vacancies formation during CSR treatment. To the best of our knowledge, most of the Co-containing layered structure cathodes utilize the $Co^{3+}/Co^{4+}$ redox couple rather than $Co^{2+}/Co^{3+}$ redox couple for charge compensation, such as the well-known layered $LiCoO_2$[55] and $LiNi_xCo_yMn_zO_2$ (NCM, $x+y+z=1$)[39,56], and such a charge compensation mechanism dominated by $Co^{2+}/Co^{3+}$ is both novel and confusing. However, the appearance of $Co^{2+}/Co^{3+}$ redox pairs is beneficial to the structural stability, because the $Co^{3+}$ cations formed at the charged state don't have a magnetic moment and serve as a buffer atom in the *TM* layer of P2-type cathode materials[56,57].

The enhanced redox activity of lattice oxygen is also responsible for the capacity recovery, as shown in Fig. 5. From the mRIXS and the corresponding integration results, we also observed asymmetry in the oxygen redox potential, in which the oxidation mainly occurred during the charging process at 4.4 to 4.8 V, while the reduction mainly occurred during the discharge process at 3.2 to 2.8 V. Recent studies on LMRO have attributed such phenomenon to the coupling of oxygen redox and cation migration and described the oxygen redox mechanism as a dynamic $(O^{2-} + TM) \rightarrow (O^- + TM_m) + e^-$, where $TM_m$ indicates a migrated *TM*[58]. The migration of *TM*s will seriously alter the electrostatic environment, and modulate the oxygen redox potential, resulting in the O projected density of states (pDOS) shifting to higher energy. Therefore, the greater voltage asymmetry of the oxygen redox behavior could relate to the migration of transition metals during the CSR process. Moreover, lower voltage oxygen redox couples with *TM*s migration in LMRO-50CS can mitigate the oxygen release and improve the reversibility associated with oxygen redox[58]. This is consistent with the integration results of mRIXS, which achieves a fully reversible oxygen redox.

## Discussion

The variation of morphology, crystal structure, surface components and valence state of *TM*s during CSR treatment were studied by SEM, ex-situ sXRD, XPS, TEM, and sXAS. Through the above analysis, we found that the CSR treatment involves the loss of Li-ion, the oxidation of the Mn from $Mn^{2+}/Mn^{3+}$ to $Mn^{4+}$ and Co from $Co^{2+}$ to $Co^{3+}$, the introduction of partial inverse spinel phase and oxygen vacancies. We have reason to speculate that the above process occurs due to the synergistic photocatalytic effect and photothermal effect, and both effects are discussed in detail in the following:

1) After electrochemical cycling, spinel structures such as $LiMn_2O_4$ and $MnO_2$ are produced on the particle surface. It has been reported that $LiMn_2O_4$ could respond to visible light[15]. Under illumination, the material can be excited to generate microsecond lifetime photo-generated hole-electron pairs to compensate for the charging process[14,15]. In addition, it is further proved that the material can respond to a longer wavelength ranging from ultraviolet to infrared, and the response is stronger under red light[15]. In addition, $MnO_2$ is a widely used photocatalyst due to its environmental friendliness, low cost, narrow bandgap, and can be excited by visible light and infrared light[16,59]. Thus, the complex surface structure composition of LMRO-50C is closely related to the CSR process. To obtain a deeper understanding of the reaction mechanism in the CSR process, we have conducted a series of studies on the optical properties of the material. The Tuac plot obtained from the ultraviolet and visible spectrophotometry diffuse reflectance spectra (UV-Vis-NIR DRS) (Supplementary Fig. 14) is shown in Fig. 6a, which indicates that the band gap energies ($E_g$) of LMRO, spinel $LiMn_2O_4$ (LMO) and LMRO-50C are 1.112 eV, 0.836 eV and 1.163 eV, respectively. The flat band potential ($E_f$) of them, calculated by electrochemical Mott-Schottky analysis are −1.065 V (LMRO), −0.924 V (LMO), −0.792 V (LMRO-50C) (*vs.* Ag/AgCl). Therefore, the conduction band potential ($E_{CB}$) of the LMRO-50C could be estimated to be 2.352 V (*vs.* Li+/Li). Based on the formula $E_{VB} = E_{CB} + E_g$, we can further obtain that the valance band potential ($E_{VB}$) of LMRO-50C is 3.515 V (*vs.* Li+/Li). The detailed analysis procedure is provided in Supplementary Note 3. With the above result as a basis, the relative potential energy diagrams of the spent electrode are reflected in Fig. 6c, where the $Co^{2+}/Co^{3+}$ and $Mn^{2+}/Mn^{4+}$ redox potentials are above the conduction band top of LMRO-50C and can therefore be oxidized, while $Ni^{2+}/Ni^{4+}$ and $Co^{3+}/Co^{4+}$, $O^{2-}/O^{n-}$ are below the valence band top and cannot be oxidized by photogenerated holes.

Based on the above results, we propose a mechanism for the photocatalytic oxidation-reduction process. The spinel phase ($LiMn_2O_4$ and $MnO_2$) components on the surface caused by electrochemical cycling absorb photo energy to generate hole and electron

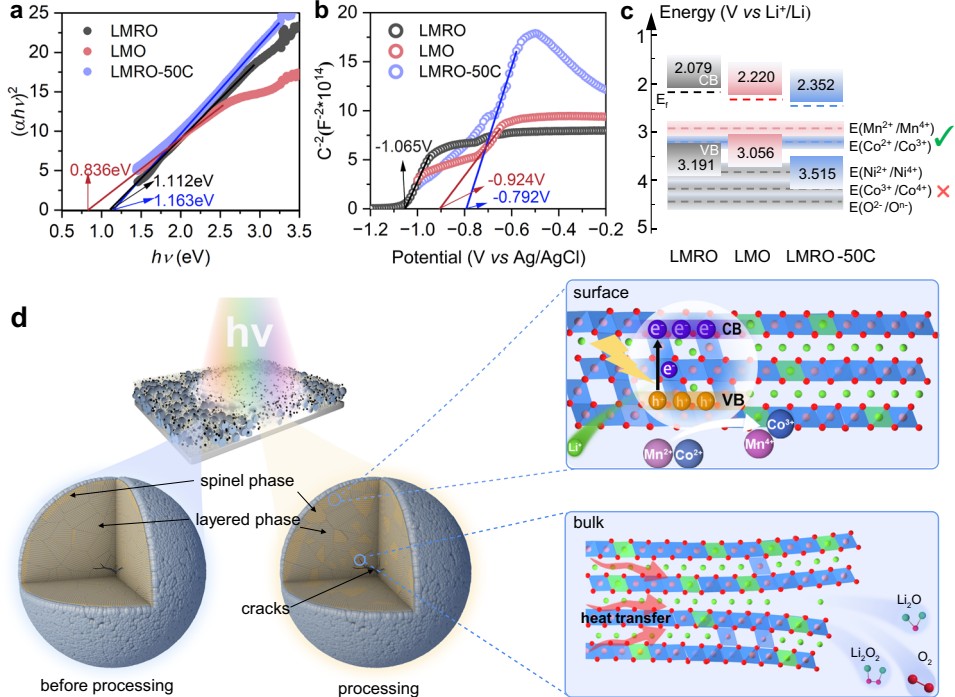

**Fig. 6 | Schematic diagram showing the principle of photocatalytic and photothermal coupling effect.** The $(\alpha h\nu)^2$ vs. $h\nu$ plots (**a**) and Mott-Schottky curves (**b**) of LMRO, LMRO-50C, LMO electrode. **c** Distribution of the conduction band potential and valence band potential of the different electrodes calculated from (**a**) and (**b**). **d** Diagram of the reaction mechanism of the CSR process.

pairs. The holes are transferred in the lattice and captured by adjacent low-valence transition metals $Mn^{2+}$, $Mn^{3+}$ and $Co^{2+}$, which are oxidized to $Mn^{4+}$ and $Co^{3+}$, respectively. As the *TM*s are oxidized, Li-ion are ejected from the structure leaving Li vacancies. Under the thermal effect generated by the CSR process, the transition metal is more likely to migrate to the li vacancies in the li layer, forming an irreversible structural phase transition. The strong (220) diffraction peak proves that the formed spinel has a partially disordered inverse spinel structure, where active Li-ion occupies the octahedral sites of manganese.

$$LiMn_2O_4 + h\nu \rightarrow LiMn_2O_4\left(h_{VB}^+ + e_{CB}^-\right) \quad (1)$$

$$MnO_2 + h\nu \rightarrow MnO_2\left(h_{VB}^+ + e_{CB}^-\right) \quad (2)$$

$$Co^{2+}, Mn^{2+}, Mn^{3+} + h_{VB}^+ \rightarrow Mn^{4+}, Co^{3+} \quad (3)$$

2) In addition to photocatalytic effects, the photothermal effect should not be overlooked for its important influence over material performance. Since CSR treatment can rapidly heat the material surface to about 350 °C, and the heat get transferred to the bulk and aggravates the cracks generated by the electrochemical cycle, leading to the limited decomposition of the material along the cracks[60]. The thermal decomposition is:

$$xLiTMO_2(defect\ larered) \rightarrow \frac{x}{4}LiMn_2O_4(spinel) + \frac{x}{2}TMO\ (rock\ salt)$$
$$+ \frac{x}{2}Li_2O + \frac{x}{2}Li_2O_2 + \frac{3x}{4}O_2\ (gas) \quad (4)$$

Under thermal effect, the defective layer structure decomposes into spinel, rock salt, Li oxides and oxygen. Various Li oxides and oxygen are transferred to the particle surface, released and evaporated.

Meanwhile, the loss of Li-ion at high temperatures above 300 °C causes the decomposition of residual peroxy-groups in the particles, producing oxides and releasing oxygen gas, which further leads to the formation of spinel structures and O vacancies:

$$O_2^{n-} \rightarrow \frac{n}{2}O^{2-} + \frac{2-n/2}{2}O_2\ (release) \quad (5)$$

In addition, oxygen vacancies that are created during cycling not only extend the photo-responsive range but also trap the electrons or holes to facilitate their combination[61,62]. Therefore, we infer that the photo-repairability of LMRO materials is related to electrodes' special degradation mechanism with the generation of oxygen vacancies and a new phase. This was confirmed by the results of our light treatment of the initial material, which did not show significant changes in the chemical valence states and electrochemical properties.

To prove the above hypothesis and compare the difference between CSR treatment and simple heat treatment, the temperature-based in-situ sXAS was performed on electrodes after 50 cycles (Supplementary Fig. 15). Based on the changes of the spectrum during the temperature rising from room temperature to 300 °C, the mean valence changes of the *TM*s are fitted, as shown in Supplementary Fig. 15d. In contrast to the rapid oxidation of $Mn^{2+}$, $Mn^{3+}$ and $Co^{2+}$ during the CSR treatment, only a small amount of Mn was oxidized, and the valence states of Co and Ni remained almost unchanged during the temperature increase to 300 °C. Therefore, the combination of rapid photocatalytic oxidation and the photothermal effect during the CSR process promotes the recovery of the electrochemical performance of the material, which has obvious advantages over conventional heat treatment.

Given the specific nature of the photocatalytic effect, the particles need to be irradiated with light of sufficient intensity, and at the appropriate wavelength for the occurrence of these CSR reactions. The porous positive electrode structure allows light irradiation on the inner particles to a certain extent, but it may also lead to a gradient

effect from the top surface to the deep bottom as the loading of the active materials increases. In addition, the effective reparability of the CSR process was shown to be diminished in LMRO samples after long cycles. The capacity of LMRO-200C was repaired after the CSR process and reached 230 mAh g$^{-1}$ (Supplementary Fig. 16), which is lower than that of LMRO-50CS. The irreparable capacity loss may come from two factors. On the one hand, it is due to the irreversible dissolution of $TM$s, especially for Mn and Ni, during the long cycling process[7,63]. On the other hand, it is attributed to the formation of irreversible rock salt structure during the long cycling process[64] and CSR process, as shown in Supplementary Figs. 17 and 18, the 3.8% and 13.6% rock salt phase weight fraction appeared in LMRO-200C and LMRO-200CS, respectively.

To summarize, in this work, cycled LMRO materials can be effectively repaired with a simple, rapid and green method (CSR treatment), and the repaired materials exhibited excellent capacity retention and voltage stability. The mechanism of capacity recovery was explored and the evolutions of the charge compensation mechanism were revealed before and after the CSR process. The sXAS and mRIXS results showed that a large amount of manganese is oxidized to tetravalent manganese during the CSR treatment process, and the charge transfer numbers of $TM$s (Mn, Co and Ni) and lattice oxygen both increase during the subsequent charge-discharge processes, resulting in capacity recovery. The improvement of electrochemical stability is related to two factors: one is the appearance of redox couples of $Co^{2+}/Co^{3+}$ and $Mn^{2+}/Mn^{4+}$ and the enhancement of redox reversibility of lattice oxygen and the other is the formation of layered-spinel (with partial inverse spinel) coherent structure after CSR treatment. We further proved that the beneficial response to the photocatalytic and photothermal effects of CSR is highly related to the formed spinel phase $LiMn_2O_4$ and $MnO_2$, and oxygen vacancy during the charge/discharge cycles. Thus, an emerging class of spinel-based high-energy-density cathode materials may be able to further optimize the electrochemical performance of the materials through post-treatment by the CSR process, such as the high-energy-density and partially ordered $Li_{1.68}Mn_{1.6}O_{3.7}F_{0.3}$ and $Li_{1.68}Mn_{1.6}O_{3.4}F_{0.6}$ materials[65]. Furthermore, the understanding of $Co^{2+}/Co^{3+}$ and $Mn^{2+}/Mn^{4+}$ redox couples will provide possibilities for the modification of LMRO materials and the design of high-energy density battery materials in the future.

## Methods

### Cathode materials synthesis
The precursor of LMRO was prepared by a co-precipitation method. The mixed transition-metal sulfate ($MnSO_4 \cdot 4H_2O$: $CoSO_4 \cdot 7H_2O$: $NiSO_4 \cdot 6H_2O = 4:1:1$ in $TM$ molar ratio) and $Na_2CO_3$ were separately dissolved in DI water to prepare solutions of 2 M. Then, the $Na_2CO_3$ solution and $NH_3 \cdot H_2O$ solution (0.2 M) were slowly dripped into the transition-metal sulfate solution in a tank reactor at 55 °C and the pH of the solution was controlled to be stable at 7.5. After stirring for 18 h the solution was vacuum-filtered, washed with DI water, and dried overnight in a vacuum oven at 100 °C. The resulting precipitates and $Li_2CO_3$ were thoroughly mixed to form the desired powder. The desired powder was heated at 500 °C for 5 h and then annealed at 850 °C for 15 h in the air, LMRO was obtained after being naturally cooled to room temperature.

### Preparation of cycled electrode
The pouch cells with LMRO as cathode and Li metal (China Energy Lithium Co., Ltd.) as anode were fabrication and cycled to obtain a cycled electrode. The electrode sheet was prepared by mixing LMRO, conductive carbon black (Super-P), and polyvinylidene fluoride (PVDF) binder with a weight ratio of 8:1:1 in N-Methyl pyrrolidone (NMP) to prepare slurry and coating it on Aluminum foil. The cathode electrode

was cut by a mold and dried in a vacuum at 120 °C for 12 h. The mass loading of the electrode was about 2.4 mg cm$^{-2}$. The pouch cell was assembled with Li metal foil as anode, Celgard as separator and 1 M $LiPF_6$ in EC: DEC = 1:1 wt.% as the electrolyte in glovebox under argon atmosphere. The cell was charged and discharged between 2.0 and 4.8 V for the initial cycle at C/10 and the rest of 50 cycles at C/2 (vs. Li$^+$/Li). After cycling, the pouch cell was disassembled in a glovebox under argon atmosphere. The cathode electrode after cycling was washed with Dimethyl carbonate (DMC) to remove residual $LiPF_6$ salts, then the electrode was dried in vacuum at 60 °C for 12 h and donated LMRO-50C.

### Reparation of cycled electrode
The cycled electrode was repaired under Concentrated solar radiation (CSR). As is shown in Fig. 1a, b, the solar simulation system with Xenon lamp light source (ABET Technologies, Model Sun 3000). Simulating solar passing through Fresnel lens matrix to obtain Focus spot matrix. Adjust the size and focus intensity of the focus spot by lifting the height of the convex lens matrix. In this experiment, the work plane is selected as the plane from the focus 5 mm, where the radiation intensity is 10941 W m$^{-2}$ and the surface temperature of the electrode can reach 350 °C. The cycled electrode moved horizontally to uniform solar irradiation treatment for 30 s and then dried for 12 h under vacuum at 120 °C. The treated sample was donated LMRO-50CS.

### Electrochemical Measurements
The repaired electrode was cut into 12 mm diameter discs as a working electrode. The electrode was assembled into a CR2032-type coin cell with Li metal as the counter electrode, Celgard 2502 as the separator and 1 M $LiPF_6$ in 1:1 (wt.%) ethylene carbonate (EC): dimethyl carbonate (DEC) as electrolyte. LMRO-50CS cells are cycled between 2.0 and 4.8 V (vs. Li$^+$/Li) with C/2 at the temperature of 30 °C by NEWARE Battery Test System (MIHW-200-160CH-B, Shenzhen, China).

Electrochemical impedance spectra (EIS) were recorded using an Ivium workstation (Ivium De zaale 11, Ivium Tech., Netherlands.).

### Ex-situ X-ray diffraction (XRD) and synchrotron XRD (sXRD)
The electrode samples were obtained by disassembling the batteries in different states in an argon-filled glovebox, and then washed with dimethyl carbonate (DMC) several times and dried in vacuum. The cathode sample was sliced into thin pieces and mounted in the hermetically sealed capillary tubes for ex-situ sXRD. Powder diffractions of all samples were taken using synchrotron XRD at the Advanced Photon Source (APS) at Argonne National Laboratory (ANL) on beamline 11-BM ($\lambda = 0.11165$ Å). The cathode samples were also measured by lab-based XRD (DX-2800 from HAOYUAN Instrument, $\lambda = 1.54186$ Å) for ex-situ characterization. Rietveld profile refinements were performed using the GSAS suite of programs.

### Morphology and surface components characterization
Morphology of particles on the electrode was acquired using field-emission scanning electron microscopy (SEM) in Molecular Foundry at 3.0 kV. The Li: Ni: Co: Mn - ratio of the materials was analyzed by inductively coupled plasma optical-emission spectrometry (ICP−OES, Thermo Fisher Scientific Co., Ltd). The surface composition analysis was performed by X-ray photoelectron spectroscopy (XPS, Thermo Scientific K-Alpha) and Raman spectra (Renishaw inVia, 532 nm).

### TEM characterization
The cross-sectional TEM specimen was prepared by focused ion beam (FIB, FEI Helios 600i) milling. To avoid sample damage by the Ga-ion

beam, a 2 μm thick Pt layer was deposited above the region of interest. The main thinning process was carried out using a 30 kV Ga-ion beam until the thickness was reduced to below 100 nm, followed by sweeping surface amorphous with a 2 kV Ga-ion beam. The high-resolution TEM image, HAADF-STEM images and EELS spectra were carried out using the Titan ETEM G2 scanning/transmission electron microscope. The optimal energy resolution of EELS was ~0.25 eV, as judged by the full-width at half-maximum of the zero-loss peak. FFT and EELS were filtered using Digital Micrograph software.

## Synchrotron sXAS and mRIXS measurements

Soft XAS (sXAS) and mRIXS measurements were performed in the endstation of Beamline 8.0.1 at the Advanced Light Source (ALS), Lawrence Berkeley National Laboratory (LBNL). The Mn $L$-edge, Ni $L$-edge, Co $L$-edge and O $K$-edge spectra were acquired at room temperature under vacuum using total electron yield (TEY) via the drain current and total fluorescence yield (TFY) via Silicon Photodiodes. All the TEY and TFY spectra were normalized to the beam flux measured by the standard sample and the resolution of the excitation energy was 0.15 eV. The mRIXS data were collected through a high-efficiency soft X-ray spectrometer with an excitation energy step size of 0.2 eV. The 2D images were obtained by normalizing and combining a series of RIXS spectra with different excitation energies. The resolution of the excitation energy and the emission energy are about 0.35 eV and about 0.25 eV, respectively.

## Time-of-flight secondary ions mass spectrometry test

The FIB-SEM/ Time-of-flight secondary-ion mass spectrometry (TOF-SIMS) was used to analyze the cross-section composition of the samples. Sample cross-section preparation and TOF-SIMS analysis were performed on a TESCAN Amber GMH SEM/FIB microscope (TESCAN CHINA, Ltd.), which is equipped with a TOF-SIMS analyzer. Ga-ion source for both FIB milling and TOF-SIMS analysis. The primary ions (Ga-ion) impact the sample surface, causing different elements and molecules to be ionized and ejected to produce characteristic secondary species (including neutral, electrons, molecules and ions). Secondary ions were collected by a mass analyzer and distinguished by the m/z calculated from the time-of-flight. Each map was obtained by scanning 50 frames at 30 KV with a beam current of 50 pA.

## In situ sXAS experiments

The in situ soft-XAS characterization was performed at beamline 13-2 has an APPES (ambient pressure photoemission spectroscopy) station at the SSRL (Stanford Synchrotron Radiation Light source), National Accelerator Laboratory. The sample temperature can be varied from 300 K to 600 K by a ceramic heater. The Mn, Ni, and Co $L$-edges were continuously collected during the heating at room temperature,100 °C, 200 °C, and 300 °C to reveal the valence state change during heating. The estimated incident X-ray energy resolution was $\Delta E/E = 1 \times 10^{-4}$ with beam size was 0.01×0.075 mm$^2$.

## Operando DEMS

The gas generation behavior of the cathode was investigated through Operando differential electrochemical mass spectroscopy (DEMS) experiments, employing a quadrupole mass spectrometer (HPR20, Hiden Analytical Ltd., UK) with a high-purity argon (99.999%) carrier gas flow rate of 2.000 mL min$^{-1}$. A cathode electrode loaded with 4.2 mg cm$^{-2}$ was assembled in a specific sealed cell mould with lithium metal as anode, Whatman as separator and 1 M LiPF6 in EC: DEC = 1:1 wt.% as electrolyte. The electrochemical behaviors of the cell were controlled by an Ivium workstation (Ivium De zaale 11, Ivium Tech., Netherlands.) at room temperature.

## Characterization of optical properties

The UV–Vis-NIR DRS was performed on a Perkin Elmer lambda 1050+ system. The Mott-Schottky was measured on Gamry interface 5000 workstations. Using a three-electrode system to obtain Mott-Schottky curves, where the electrode sheet as the working electrode, the platinum sheet as the counter electrode and the Saturated KCl/Ag/AgCl electrode as a reference electrode. The electrolyte was 0.2 M Na$_2$SO$_4$ aqueous solution and purged with nitrogen gas for one hour before measurement. Seal the current collector and the connection between the current collector and the electrode with epoxy resin to ensure that only the electrode material is immersed in Na$_2$SO$_4$ electrolyte. Mott-Schottky measurements were performed the test potential ranged from 0.5 V to −1.0 V (*vs.* Ag/AgCl) and the frequency was kept constant at 1000 Hz.

## Data availability

The data that support the findings of this study are available within the paper and its Supplementary Information and all data are also available from the corresponding authors upon reasonable request. Source data are provided with this paper.

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

## Acknowledgements

This work was supported by the Shenzhen Stable Support Plan Program for Higher Education Institutions Research Program (20220815153728002, Y.J.L.).

## Author contributions

X.H. conceived and designed this experiment. X.H., Y.J.L. supervised the research. H.L.W. performed experiment and initial data analysis and wrote the manuscript. X.G. and L.Y.H. conducted the design and construction of experimental device for ex-situ sXRD and sXAS. H.L.W. and Y.D.Z. proceeded FIB preparation and conducted the TEM and STEM-EELS measurement and analysis. J.W. instructed the DEMS experiment operation and data analysis. J.L., Y.K.X., Z.M.L. and J.W. joined the scientific discussion of data and revised the manuscript. All authors contributed to the discussion and provided feedback on the manuscript. All authors contributed to the interpretation, conclusions, and preparation of the manuscript.

## Competing interests

The authors declare no competing interests.
