## [Peer Review File · Nature Communications]

REVIEWER COMMENTS

Reviewer #1 (Remarks to the Author):

The authors in this manuscript report a concentrated solar radiation (CSR) strategy for direct recycling of LMRO electrode, which enables the recovery of capacity and effectively improves its electrochemical stability. The cathode electrolyte interface (CEI) on surface and metastable state structure of cycled material provides the precondition for photocatalytic reaction and thermal reconstruction during CSR processing. The corresponding mechanism was investigated in details by various characterization methods. The concept is novel and the manuscript is overall well presented. To enhance the presentation, there are some major concerns from the reviewers as follows.

1. After the LMRO cathode particles after 50 cycles of CSR treatment are mentioned in the article, it can be clearly seen in the SEM diagram that the CEI film on the original surface is partially removed, so what are the main components of the CEI film that are partially removed? In what way or what reaction did it take off the electrode surface? Can the authors give more explanation.
2. Comparing the XAS spectra of LMRO-Pristine, LMRO-50C and LMRO-50CS samples given in the manuscript, it can be seen that only the peak strength and peak position of the i peak of Mn have changed, while the other two TM elements only have the change of peak strength, corresponding explanations are required.
3. The manuscript proposes that when LMRO-50CS is charged and discharged in the voltage range of 2.8V-4.8V, lattice oxygen can also be used as an active element in the high and low voltage range to participate in the redox reaction of charge compensation, which is a relatively innovative statement. Because in most cathode materials we usually need to avoid lattice oxygen from participating in the reaction, and it is difficult for oxygen generated by lattice oxygen oxidation to re-enter the lattice. Can the authors supplement the experimental data measuring the gas production of the battery during the charging and discharging process to support the high reversibility of the lattice oxygen redox reaction?
4. From the XRD, SEM, TEM and other experimental data provided in the paper, it can be seen that the structure of the LMRO-50CS sample has undergone great changes, and many irreversible spinel phases have appeared, and the authors explain that when TMs are oxidized, lithium ions are delithified from the structure, leaving lithium vacancies. Under the thermal effect generated by light, transition metals are more likely to migrate to lithium vacancies in the lithium layer, forming irreversible structural phase transitions. Although the oxidation of TMs is the desired outcome, does the accompanying spinel phase impede the diffusion process of Li^+ ? At the same time, in terms of electrochemical performance, can the battery still have better capacity at high rate conditions?
5. This work mainly presents the conceptual methodology with a thin cathode casting (2.4 mg cm^{-2}). The reviewer is curious about the penetrability of incident radiation subject to thick electrode. If only the surface layer of electrode can be directly regenerated via photocatalysis, the bottom counterpart that accounts for the majority cannot be recycled. The limitations of this work also need to be further discussed.

Reviewer #2 (Remarks to the Author):

This manuscript presents direct recycling strategies for lithium- and manganese-rich layered oxides (LMRO) utilizing concentrated solar radiation (CSR). The application of photocatalytic and photothermal effects during CSR treatment facilitates the recovery of spinel LiMn_2O_4 and MnO_2 during the charge/discharge cycle and the cathode electrolyte interface (CEI) on the surface. The reaction mechanisms of CSR treatment on the LMRO electrode are comprehensively proposed based on state-of-the-art analysis. While the direct recycling method for LMRO is intriguing in terms of capacity recovery and enhanced electrochemical stability, systematic characterization and further development of this strategy are imperative before its publication in Nature Communications. Here are the reviewer's comments:

(1) The primary interest in LMRO stems from high-energy density. Although the CSR treatment recovered capacity and mitigated voltage decay, the average discharge voltage of LMRO-50CS was notably low, undermining the advantages of LMRO (Figure 3d). If the authors claim complete performance restoration, they should provide a comparison of energy density between LMRO and LMRO-50CS. Furthermore, while both LMRO and LMRO-50CS displayed nearly constant voltage hysteresis from cycles 1 to 150, the early cycle voltage hysteresis of LMRO-50CS (1st-50th cycle) was more pronounced than that of LMRO. This raises concerns about the high-energy density cathode benefits of LMRO-50CS. Additionally, introducing a spinel phase with three-dimensional channels via CSR might enhance LMRO's diffusion properties. However, this manuscript does not present details of LMRO's rate capability.

(2) The manuscript claims that the structural distortion in LMRO-50CS was effectively suppressed due to a smaller shift in the (003) peak compared to LMRO in ex-situ XRD (Figure 3e). However, the substantial difference in lithium content between LMRO-50CS's 1st cycle ($\text{Li}_{0.762}\text{Mn}_{0.56}\text{Ni}_{0.13}\text{Co}_{0.11}\text{O}_2$) and LMRO's 50th cycle ($\text{Li}_{1.014}\text{Mn}_{0.56}\text{Ni}_{0.15}\text{Co}_{0.13}\text{O}_2$) was overlooked, which may potentially distort the truth. To address this, the reviewer suggests presenting ex-situ XRD data for the LMRO-50CS 2nd cycle, which has a larger lithium content and can serve as a genuine control group. It might be more informative to perform a quantitative analysis by expressing the stress-strain curve through Rietveld refinement rather than using the shift of the (003) peak for a qualitative analysis.

(3) The authors claimed that the precondition for CSR is the electrochemically formed spinel LiMn_2O_4 and MnO_2 , which generate hole and electron pairs. This mechanism could be extended not only to LMRO recovery but also to pristine spinel-phase-based Li-rich cathode materials, e.g., $\text{Li}_{1.68}\text{Mn}_{1.60}\text{O}_{3.7}\text{F}_{0.3}$ and $\text{Li}_{1.68}\text{Mn}_{1.60}\text{O}_{3.4}\text{F}_{0.6}$ by Ceder's group (DOI: 10.1038/s41560-020-0573-1 written by Huiwen Ji et al.). By utilizing the CSR treatment for both the development of pristine materials and capacity recovery, the value of the authors' work can be enhanced. In this context, please provide the authors' perspective on the manuscript.

(4) LMRO-50CS contained the spinel-layered coherent phase with a partially inverted spinel structure due to photocatalytic and photothermal effects from the electrochemically formed spinel phase and defect layered structure. However, the important missing point was the existence of the rock salt phase, which mainly associated with the inactive phase of lithium-rich layered oxides. According to the CSR mechanism, rock salt phase recovery was limited, and its formation was enhanced by photothermal effects. " $x\text{LiTMO}_2$ (defect layered) $\rightarrow x/4 \text{Li} [\text{Mn}]_2\text{O}_4$ (spinel) + $x/2 \text{TMO}$ (rock salt) + $x/2 [\text{Li}]_2\text{O} + x/2 [\text{Li}]_2\text{O}_2 + 3x/4 \text{O}_2$ (gas)". In this regard, there is concern about the effectiveness of CSR

when LMRO contains a larger portion of the rock salt phase. Electrochemical tests and XRD data for LMRO-200CS (LMRO 200cycles followed by the CSR treatment) should be provided.

(5) Following the previous question, please elaborate on how the photothermal effect reaction represents Li₂O and Li₂O₂ products and their roles. Experimental data concerning these products are missing.

(6) The claim that CSR partially removed the dense CEI film of electrochemical decomposition products without affecting binders and conductive carbon was somewhat contradictory. First of all, the reviewer cannot find the morphological changes of electrode before and after the CSR treatment (Figure 1c-d and supplementary Fig.2). To substantiate the claim, a comparison of CEI components on the electrode surface through XPS analysis was more compelling than vague SEM images. If we assume that CEI is decomposed by CSR, the decomposition of binders should be considered based on the TGA-DTA results. In supplementary Fig. 3 (TGA-DTA data), the decomposition temperature of CEI was around 340oC, while that of binder was about 300oC. Considering the weight reduction at the binder decomposition temperature in TGA-DTA of LMRO-50CS was less than that of LMRO-Pristine, substantial binder decomposition during CSR treatment is doubtful. Moreover, according to TOF-SIMS analysis (supplementary Fig. 4), it seems that CSR induces cracks leading to severe deterioration at high voltages.

Minor rectifications:

(1) For clarity, legends for Fig. 4 and 5 should be made clearer. Annotate corresponding voltages with charge or discharge states, for instance: 50CS_3.2V → 50CS_3.2V (Discharge), 50CS_4.8V → 50CS_4.8V (Charge)

(2) Please add the notation "O K-edge" to the EELS data in Fig. 5a-b.

(3) Correct the misspellings: use "CSR" instead of "FSI" and "LMRO" rather than "LRMO" in Supplementary Fig.3.

REVIEWER COMMENTS

Point-by-point response to the comments

We would like to thank Reviewers for their constructive criticism, comments and suggestions. Please find below our point-by-point responses to the Reviewers' comments.

Response to Reviewer 1# (Remarks to the Authors) :

The authors in this manuscript report a concentrated solar radiation (CSR) strategy for direct recycling of LMRO electrode, which enables the recovery of capacity and effectively improves its electrochemical stability. The cathode electrolyte interface (CEI) on surface and metastable state structure of cycled material provides the precondition for photocatalytic reaction and thermal reconstruction during CSR processing. The corresponding mechanism was investigated in details by various characterization methods. The concept is novel and the manuscript is overall well presented. To enhance the presentation, there are some major concerns from the reviewers as follows.

1. After the LMRO cathode particles after 50 cycles of CSR treatment are mentioned in the article, it can be clearly seen in the SEM diagram that the CEI film on the original surface is partially removed, so what are the main components of the CEI film that are partially removed? In what way or what reaction did it take off the electrode surface? Can the authors give more explanation.

Answers: Thanks for the suggestions from the reviewer. In order to investigate what components are partially removed in the CEI layer from the CSR process, XPS characterization has been performed for the LMRO-Pristine, LMRO-50C and LMRO-50CS samples. The XPS spectra and fitting for each corresponding species are shown in Fig. 2a-d. Based on the fitting results of XPS we can deduce that the organic components like $\text{ROCO}_2\text{-Li}$, R-CF_x , and $\text{R-CO}_x\text{F}_y$, polycarbonates in CEI layer are almost completely removed from the CEI. As the CSR process causes localized high temperatures of up to 350 °C, the organic components of the CEI are mainly removed by thermal decomposition, due to their poor thermal stability [InfoMat, (2021), 648-661, 3(6)]. The specific text in line 147-165 and Fig. 2a-d in the revised manuscript are also attached below.

“To elucidate the mechanism of reactions occurred on CEI layer during the CSR process, X-ray photoelectron spectroscopy (XPS) spectra are collected on LMRO-Pristine, LMRO-50C and LMRO-50CS, as is shown in Fig. 2a-d. All three samples show a strong C-C peak located at 284.8 eV, which is attributed to the conductive carbon black (super-P) in electrode. LMRO-50C sample shows stronger C-O and C=O peaks in C 1s spectra at 286.6 eV and 288.8 eV and in O 1s at 533.6 eV and 532.6 eV compared with LMRO-pristine, which attribute to organic components like $\text{ROCO}_2\text{-Li}$, R-CF_x , and $\text{R-CO}_x\text{F}_y$, polycarbonates in CEI layer formed during cycling [Nature

Communications, (2020), 3629, 11(1); *Matter*, (2021), 302-312, 4(1); *Nature Communications*, (2022), 6966, 13(1)]. In F 1s spectra, a relatively high proportion of inorganic components, including Li_xPF_y (686.8 eV), $\text{Li}_x\text{PF}_y\text{O}_z$ (685.6 eV), and LiF (684.7 eV), are observed on the LMRO-50C surface, which attribute to the decomposition of LiPF_6 salt [*Adv. Energy Mater.* (2018), 8, 1801957;]. The enrichment of LiF (55.8 eV) and Li_2CO_3 (55.2 eV) in CEI appears in the Li 1s spectrum as well, which agrees with findings in previous report [*Journal of Power Sources*, (2016), 31-40, 329; *Matter*, (2021), 302-312, 4(1)]. In comparison with LMRO-50C, the C-O and C=O peaks observed in the C 1s and O 1s spectra nearly vanished in the LMRO-50CS sample. This disappearance can be attributed to the decomposition of organic components within the CEI, induced by localized high temperatures during the CSR process. It is noteworthy that the peaks representing the inorganic components in the CEI layer do not change significantly due to the better thermal stability of these inorganic components [*Nature Communications*, (2020), 3629, 11(1); *InfoMat*, (2021), 648-661, 3(6)]. As a result, the CEI layer of LMRO-50CS sample become rich in inorganic components like LiF, Li_2CO_3 , Li_xPF_y and $\text{Li}_x\text{PF}_y\text{O}_z$, which contribute to the improved cycling stability of the repaired material. [*InfoMat*, (2021), 648-661, 3(6); *Energy Storage Materials*, (2021), 77-86, 37; *Advanced Functional Materials*, (2023), 33(19)]”

Fig. 2a-d Surface composition changes from LMRO to LMRO-50C to LMRO-50CS. XPS spectra of C 1s (a), O 1s (b), F 1s (c), Li 1s (d)

2. Comparing the XAS spectra of LMRO-Pristine, LMRO-50C and LMRO-50CS samples given in the manuscript, it can be seen that only the peak strength and peak position of the i peak of Mn have changed, while the other two TM elements only have the change of peak strength, corresponding explanations are required.

Answers: Thanks for the reviewer’s comment. Here in the sXAS spectra of LMRO-Pristine, LMRO-50C and LMRO-50CS samples, the difference is quite obvious for the sample of LMRO-50C at 2.0V and this electrode after CSR process, we agree with the reviewer that the peak strength and peak position of the i peak of Mn in Fig. 2e have

changed, while the other two *TM* elements (Co and Ni) only have the change of peak strength. For the sXAS of *TM* L-edges, the peak shapes, peak offsets, and peak intensities are closely related to the valence states of the corresponding elements. By fitting the standard spectra using Linear combination fitting, we calculated the elemental valence states at different states. From LMRO-Pristine to LMRO-50C to LMRO-50CS, the valence state of Mn undergoes a dramatic change from 3.68+ to 2.77+ to 3.81+. However, for Co and Ni, the valence changes that undergo three states are from 2.91+ to 2.85+ to 2.93+ and from 2.33+ to 2.06+ to 2.05+, respectively. As a result, the relatively small range of valence changes in Co and Ni causes only slight changes for peak strength in the spectra.

3. The manuscript proposes that when LMRO-50CS is charged and discharged in the voltage range of 2.8V-4.8V, lattice oxygen can also be used as an active element in the high and low voltage range to participate in the redox reaction of charge compensation, which is a relatively innovative statement. Because in most cathode materials we usually need to avoid lattice oxygen from participating in the reaction, and it is difficult for oxygen generated by lattice oxygen oxidation to re-enter the lattice. Can the authors supplement the experimental data measuring the gas production of the battery during the charging and discharging process to support the high reversibility of the lattice oxygen redox reaction?

Answer: Thanks for the comments. We agree with the review that most cathode materials that we usually used need to avoid the release of oxygen, which formed from the fully oxidation of lattice oxygen. Here in this work, higher proportion of oxygen defects on the surface of LMRO-50CS lead to a well-controlled oxygen oxidation, which has been reported in refs [Journal of Materials Chemistry A, (2020), 8(16), 7733-7745; Energy and Environmental Science, (2022), 15(10), 4137-4147; Advanced Functional Materials, (2023), 33(18), 2214613]. Therefore, the long plateau at 4.6V is shorted and no oxygen gas release occur for the LMRO-50CS sample during the electrochemical process for the first cycle. In order to support the conclusion for the high reversibility of the lattice oxygen redox reaction, DEMS measurement is performed and the result (as is shown in Supplementary Fig. 13) demonstrated that no oxygen is released during the first charging process. The Supplementary Fig. 13 in the revised manuscript is attached below.

Supplementary Fig. 13 Operando DEMS profile of LMRO-50CS performed during the first charging process to 4.8 V (vs. Li^+/Li)

4. From the XRD, SEM, TEM and other experimental data provided in the paper, it can be seen that the structure of the LMRO-50CS sample has undergone great changes, and many irreversible spinel phases have appeared, and the authors explain that when TMs are oxidized, lithium ions are delithiated from the structure, leaving lithium vacancies. Under the thermal effect generated by light, transition metals are more likely to migrate to lithium vacancies in the lithium layer, forming irreversible structural phase transitions. Although the oxidation of TMs is the desired outcome, does the accompanying spinel phase impede the diffusion process of Li^+ ? At the same time, in terms of electrochemical performance, can the battery still have better capacity at high-rate conditions?

Answer: Thanks for the questions from the reviewer. In order to investigate the influence on the Li -ions diffusion for the appearance of spinel phase, we measure the rate performance of LMRO-50CS, as shown in Fig. 3d and Supplementary Fig. 10 in revised version of manuscript. The LMRO-50CS delivers a higher discharge capacity at 0.1C, whereas the discharge capacity at 10C is slightly lower than LMRO. Therefore, it can be concluded that the existence inverse spinel phase in LMRO-50CS particle indeed slightly limited the diffusion process of Li -ions. The Li -ion diffusion coefficients of LMRO and LMRO-50CS are $D_{\text{Li}^+}(\text{LMRO}) = 6.868 \times 10^{-17} \text{cm}^2 \text{S}^{-1}$ and $D_{\text{Li}^+}(\text{LMRO-50CS}) = 1.936 \times 10^{-17} \text{cm}^2 \text{S}^{-1}$, calculated from the EIS measurement, which also proves that the diffusion of li -ion is slightly impeded. The lower D_{Li^+} may be due to the longer Li -ion diffusion path in the layered-spinel coherent structure, where Li -ion need to pass through multiple layered grains and defects to reach the particle surface. However, the contribution of this layered-spinel coherent structure on maintaining cycling stability is more obvious. To make it more intuitive for you to review, Fig. 3d and Supplementary Fig. 10 in the revised manuscript are attached below.

Fig. 3d Rate performance of LMRO and LMRO-50CS.

Supplementary Fig. 10 Galvanostatic charge–discharge performance of LMRO (a) and LMRO-50CS (b) at C rates of C/10, C/5, C/2, 1C, 2C, 5C and 10C, respectively (1C = 250 mAh g⁻¹). Nyquist plots (c) and profiles of Z'' vs $\omega^{-1/2}$ (d) of the LMRO and LMRO-50CS before cycle.

5. This work mainly presents the conceptual methodology with a thin cathode casting (2.4 mg cm⁻²). The reviewer is curious about the penetrability of incident radiation subject to thick electrode. If only the surface layer of electrode can be directly regenerated via photocatalysis, the bottom counterpart that accounts for the majority cannot be recycled. The limitations of this work also need to be further discussed.

Answer: Thanks for the reminding from the reviewer. We agree with the reviewer that the limitation of this CSR treatment is mainly from the thickness of electrode. There will be a gradient for the regeneration effect for particles from the top to the bottom of electrode. Here in this work, we demonstrate that the CSR technique could successfully repairs the electrode and result a fully recovered capacity with long-term cycling stability. Therefore, we would like to highlight the scientific concept of this work, whereas the repairing mechanism and the corresponding reaction mechanism of LMRO-50CS during the electrochemical process is clearly revealed. In addition, we have added relevant discussions on the limitations of this work in line 460-464 in the revised manuscript and also attached below.

“Given the specific nature of the photocatalytic effect, the particles need to be irradiated with light of sufficient intensity, and at the appropriate wavelength for the occurrence of these CSR reactions. The porous positive electrode structure allows light irradiation on the inner particles to a certain extent, but it may also lead to a gradient effect from the top surface to the deep bottom as the loading of the active material increasing.”

Response to Reviewer 2# (Remarks to the Authors) :

This manuscript presents direct recycling strategies for lithium- and manganese-rich layered oxides (LMRO) utilizing concentrated solar radiation (CSR). The application of photocatalytic and photothermal effects during CSR treatment facilitates the recovery of spinel LiMn₂O₄ and MnO₂ during the charge/discharge cycle and the cathode electrolyte interface (CEI) on the surface. The reaction mechanisms of CSR treatment on the LMRO electrode are comprehensively proposed based on state-of-the-art analysis. While the direct recycling method for LMRO is intriguing in terms of capacity recovery and enhanced electrochemical stability, systematic characterization and further development of this strategy are imperative before its publication in Nature Communications. Here are the reviewer's comments:

1. The primary interest in LMRO stems from high-energy density. Although the CSR treatment recovered capacity and mitigated voltage decay, the average discharge voltage of LMRO-50CS was notably low, undermining the advantages of LMRO (Figure 3d). If the authors claim complete performance restoration, they should provide a comparison of energy density between LMRO and LMRO-50CS. Furthermore, while both LMRO and LMRO-50CS displayed nearly constant voltage hysteresis from cycles 1 to 150, the early cycle voltage hysteresis of LMRO-50CS (1st-50th cycle) was more pronounced than that of LMRO. This raises concerns about the high-energy density cathode benefits of LMRO-50CS. Additionally, introducing a spinel phase with three-dimensional channels via CSR might enhance LMRO's diffusion properties. However, this manuscript does not present details of LMRO's rate capability.

Answer: Thanks for the reviewer's comments. Here in the manuscript, we describe that the LMRO-50CS enables a recovery of capacity rather than the energy density. Considering on the point for the restoration of energy density, the comparison between LMRO and LMRO-50CS in Supplementary Fig. 9d and attached below. Though the energy density of LMRO-50CS is slight lower than the LMRO sample in initial cycles, it can be well-maintained during the cycling and demonstrates a higher energy density as reaching to 6th cycle. Indeed, the induce of inverse spinel phase by CSR process slightly lower the diffusion of Li-ion in host structure, the influence become obvious under high-rate. In order to present the performance of LMRO and LMRO-50CS, the rate capability of these samples is shown in Fig. 3d and Supplementary Fig. 10 in revised manuscript. Furthermore, the Li-ion diffusion coefficient (D_{Li^+}) of LMRO-50CS obtained from the EIS spectra is $1.936 \times 10^{-17} cm^2 S^{-1}$, which is slightly smaller than the value of $6.868 \times 10^{-17} cm^2 S^{-1}$ for the LMRO sample. The lower D_{Li^+} may be due to the longer Li-ion diffusion path in the layered-spinel coherent structure, where Li-ion need to pass through multiple layered grains and defects to reach the particle surface. However, the contribution of this layered-spinel coherent structure on maintaining cycling stability is more obvious. The Supplementary Fig. 9d, and Fig. 3d and Supplementary Fig. 10 in revised manuscript are attached below.

Supplementary Fig. 9d Discharge energy density of LMRO and LMRO-50CS at 0.5C.

Fig. 3d Rate performance of LMRO and LMRO-50CS.

Supplementary Fig. 10 Galvanostatic charge–discharge performance of LMRO (a) and LMRO-50CS (b) at C rates of C/10, C/5, C/2, 1C, 2C, 5C and 10C, respectively (1C = 250 mAh g⁻¹). Nyquist plots (c) and profiles of Z' vs $\omega^{-1/2}$ (d) of the LMRO and LMRO-50CS before cycle.

2. The manuscript claims that the structural distortion in LMRO-50CS was effectively suppressed due to a smaller shift in the (003) peak compared to LMRO in ex-situ XRD (Figure 3e). However, the substantial difference in lithium content between LMRO-50CS's 1st cycle ($\text{Li}_{0.762}\text{Mn}_{0.56}\text{Ni}_{0.13}\text{Co}_{0.11}\text{O}_2$) and LMRO's 50th cycle ($\text{Li}_{1.014}\text{Mn}_{0.56}\text{Ni}_{0.15}\text{Co}_{0.13}\text{O}_2$) was overlooked, which may potentially distort the truth. To address this, the reviewer suggests presenting ex-situ XRD data for the LMRO-50CS 2nd cycle, which has a larger lithium content and can serve as a genuine control group. It might be more informative to perform a quantitative analysis by expressing the stress-strain curve through Rietveld refinement rather than using the shift of the (003) peak for a qualitative analysis.

Answer: As suggested by the reviewer, the ex-situ XRD spectra for the 2nd cycle of LMRO-50C and LMRO-50CS sample are updated in Supplementary Fig. 11 in revised manuscript. The content of Li-ions in host structure are relatively the same for both samples, but the shift of (003) peak for LMRO-50CS is still smaller compare to LMRO-50C. Therefore, it can be concluded that the variation on the lattice size is smaller for LMRO-50CS during Li-ion insertion and extraction, which leads to a more limited strain-stress level through the particle. Besides the characterization for the 2nd cycle of

(003) peak shift by ex-situ XRD, the lattice parameters through Rietveld refinement are shown in Supplementary Fig. 11b and detailed in Supplementary Table 2, which provides a quantitative analysis of the evolution of the crystal structure. As is shown in Supplementary 11b, the evolution of the crystal structure also proves a reduced proportion on the crystal size changes for LMRO-50CS during the electrochemical process.

Supplementary Fig. 11 and Table 2 in the revised manuscript are attached below:

Supplementary Fig. 11 a Ex-situ X-ray powder diffraction (XRD) of LMRO-50C and LMRO-50CS samples at representative voltage with the enlarged view in the range of $2\theta = 17.5-19.5^\circ$ (i) and $25-35^\circ$ (ii), and the corresponding selected voltage stages in the second cycle profiles. **b** The corresponding lattice parameters a and c at different electrochemical states.

Supplementary Table 2. Refined values of the lattice parameters

Samples		a/b (Å)	c (Å)	V (Å ³)	Rwp	Rp
LMRO-50C	a (CC to 3.6V)	2.856656± 0.000062	14.342482± 0.000506	101.361± 0.005	4.73	3.6
	b (CC to 4.4V)	2.846077± 0.000108	14.416471± 0.000841	101.131± 0.011	3.46	2.74
	c (CC to 4.8V)	2.842450± 0.000076	14.410069± 0.000713	100.828± 0.007	5.61	4.47
	d (DC to 4.0V)	2.843573± 0.000091	14.398351± 0.000937	100.826± 0.009	4.36	3.48
	e (DC to 3.2V)	2.864868± 0.000067	14.334563± 0.000494	101.888± 0.005	4.55	3.52
	f (DC to 2.0V)	2.86392± 0.000050	14.323689± 0.000412	102.133± 0.004	4.59	3.62
LMRO-50CS	a' (CC to 3.6V)	2.853625± 0.000092	14.220580± 0.000571	100.286± 0.014	4.94	3.64
	b' (CC to 4.4V)	2.865316± 0.000124	14.239523± 0.000551	101.244± 0.008	5.54	4.37
	c' (CC to 4.8V)	2.855927± 0.000175	14.252447± 0.001397	100.673± 0.014	4.96	3.94
	d' (DC to 4.0V)	2.873508± 0.000153	14.253320± 0.000956	102.988± 0.026	4.95	3.65
	e' (DC to 3.2V)	2.870348± 0.000181	14.273851± 0.000873	101.845± 0.012	4.65	3.69
	f' (DC to 2.0V)	2.853359± 0.000194	14.218466± 0.001170	100.253± 0.017	4.29	3.37

3. The authors claimed that the precondition for CSR is the electrochemically formed spinel LiMn₂O₄ and MnO₂, which generate hole and electron pairs. This mechanism could be extended not only to LMRO recovery but also to pristine spinel-phase-based Li-rich cathode materials, e.g., Li_{1.68}Mn_{1.60}O_{3.7}F_{0.3} and Li_{1.68}Mn_{1.60}O_{3.4}F_{0.6} by Ceder's group (DOI: 10.1038/s41560-0200573-1 written by Huiwen Ji et al.). By utilizing the CSR treatment for both the development of pristine materials and capacity recovery, the value of the authors' work can be enhanced. In this context, please provide the authors' perspective on the manuscript.

Answer: Thanks for kind reminding from reviewer. We fully agree with the reviewer that this CSR treatment may also work for other spinel-phase-based Li-rich cathode materials. There are three fundamental requirements for the occurrence of CSR process, including the existence of spinel phase, the oxidation state of Mn should be lower than 4+. Since the oxidation state of Mn in spinel-phase-based Li_{1.68}Mn_{1.60}O_{3.7}F_{0.3} and Li_{1.68}Mn_{1.60}O_{3.4}F_{0.6} is 3.76+ and 3.58+, it is possible for the development and capacity recovery of these materials by utilizing the CSR treatment. In order to enhance the value of this work, we have cited this reference and provide a perspective in line 483-489 in the revised manuscript. The specific context is also attached below.

"We further proved that beneficial response to the photocatalytic and photothermal

effects of CSR is highly related to the formed spinel phase LiMn_2O_4 and MnO_2 , and oxygen vacancy during the charge/discharge cycles. Thus, an emerging class of spinel-based high-energy-density cathode materials may be able to further optimize the electrochemical performance of the materials through post-treatment by the CSR process, such as the high-energy-density and partially ordered $\text{Li}_{1.68}\text{Mn}_{1.60}\text{O}_{3.7}\text{F}_{0.3}$ and $\text{Li}_{1.68}\text{Mn}_{1.60}\text{O}_{3.4}\text{F}_{0.6}$ materials [Nature Energy, (2020), 213-221, 5(3)].

4. LMRO-50CS contained the spinel-layered coherent phase with a partially inverted spinel structure due to photocatalytic and photothermal effects from the electrochemically formed spinel phase and defect layered structure. However, the important missing point was the existence of the rock salt phase, which mainly associated with the inactive phase of lithium-rich layered oxides. According to the CSR mechanism, rock salt phase recovery was limited, and its formation was enhanced by photothermal effects. “ $x\text{LiTMO}_2$ (defect layered) $\rightarrow x/4 \text{Li} [\text{Mn}]_2 \text{O}_4$ (spinel) + $x/2 \text{TMO}$ (rock salt) + $x/2 [\text{Li}]_2 \text{O}$ + $x/2 [\text{Li}]_2 \text{O}_2 + 3x/4 \text{O}_2$ (gas)”. In this regard, there is concern about the effectiveness of CSR when LMRO contains a larger portion of the rock salt phase. Electrochemical tests and XRD data for LMRO-200CS (LMRO 200cycles followed by the CSR treatment) should be provided.

Answer: Thanks for the comments from the reviewer. As suggested by the reviewer, the electrochemical tests and XRD data for LMRO-200CS (LMRO 200cycles followed by the CSR treatment) are provided in Supplementary Fig. 16 and Supplementary Fig. 18 in revised manuscript. As shown in the XRD spectra in Supplementary Fig. 17 and 18, the weight fraction of rock salt phase is 3.8% in the LMRO-200C and 13.6% in LMRO-200CS sample, so the effect of CSR treatment on capacity recovery is undermined. The discharge capacity can only reach to $230 \text{ mAh} \cdot \text{g}^{-1}$ at C/2 for the LMRO-200CS sample, as shown in the electrochemical profiles in Supplementary Fig. 16. Though the effectiveness capacity recovery is reduced due to the existence of larger portion of rock salt phase and dissolution of TMs, the cycling stability can be still well-maintained. We have discussed this situation in line 464-472 in the revised manuscript. For your review, the specific text and corresponding figures in revised manuscript are attached below.

“In addition, the effective reparability of the CSR process was shown to be diminished in LMRO samples after long cycles. The capacity of LMRO-200C was repaired after the CSR process and reached $230 \text{ mAh} \cdot \text{g}^{-1}$ (Supplementary Fig. 16), which is still lower than that of LMRO-50CS. The irreparable capacity loss may come from two factors. On the one hand, it is due to the irreversible dissolution of TMs, especially for Mn and Ni, during the long cycling process [Nature communications, 2014, 5(1): 3529. Advanced Energy Materials, 2020, 10(41): 2002506.; Small, 2023: 2301834.]. On the other hand, it is attributed to the formation of irreversible rock salt structure during the long cycling process [Physical Chemistry Chemical Physics, 2012, 14(37): 12875-12883.; Nature, 2022, 606(7913): 305-312.] and CSR process, as shown in Supplementary Fig. 17 and Supplementary Fig. 18, the 3.8% and 13.6% rock salt phase

weight fraction appeared in LMRO-200C and LMRO-200CS, respectively.”

Supplementary Fig. 16. Cycle performance of LMRO, LMRO-200C and LMRO-200CS.

Supplementary Fig. 17 Refined X-ray powder diffraction pattern of LMRO-200C.

Supplementary Fig. 18 Refined X-ray powder diffraction pattern of LMRO-200CS.

5. Following the previous question, please elaborate on how the photothermal effect reaction represents Li_2O and Li_2O_2 products and their roles. Experimental data concerning these products are missing.

Answer: Thanks for the comment from the reviewer. During the CSR process, Li and O may extract in the form of Li_2O and Li_2O_2 from the host material, this reaction has been discussed in the ref. From the TEM image and Raman spectra (shown in Supplementary Fig. 6) of resulted LRMO-50CS sample, there is no crystal lattice or the representative peaks of Li_2O and Li_2O_2 can be identified. Therefore, we come to the conclusion that the Li_2O and Li_2O_2 are formed as intermedia products, which are further removed at the end of CSR process.

6. The claim that CSR partially removed the dense CEI film of electrochemical decomposition products without affecting binders and conductive carbon was somewhat contradictory. First of all, the reviewer cannot find the morphological changes of electrode before and after the CSR treatment (Figure 1c-d and supplementary Fig.2). To substantiate the claim, a comparison of CEI components on the electrode surface through XPS analysis was more compelling than vague SEM images. If we assume that CEI is decomposed by CSR, the decomposition of binders should be considered based on the TGA-DTA results. In supplementary Fig. 3 (TGA-DTA data), the decomposition temperature of CEI was around 340°C , while that of binder was about 300°C . Considering the weight reduction at the binder decomposition temperature in TGA-DTA of LMRO50CS was less than that of LMRO-Pristine, substantial binder decomposition during CSR treatment is doubtful. Moreover, according to TOF-SIMS analysis (supplementary Fig. 4), it seems that CSR induces cracks leading to severe deterioration at high voltages.

Answer: Thanks for the comments from the reviewer. As shown in Fig. 1c-d and supplementary Fig. 2, it can be observed that there are no obvious morphology changes for electrode after CSR treatment, whereas only part of components on the surface are removed. We appreciate for the suggestion of XPS measurement, which quite helpful on the analysis for components on the surface of electrode (as shown in Fig. 2a-d). Based on the XPS results we can conclude that the organic components in the CEI layer decompose under the photothermal effect, while the inorganic components are enriched on the particle surface. TG-DTG curves are shown in Supplementary Fig. 8 of the revised manuscript, showing a small amount of decomposition of PVDF binder. However, CSR is a very fast process where the electrode sheet is exposed to concentrated solar light for only 30s, which effectively prevents a large amount of PVDF decomposition. The XPS spectra of the F1s side also effectively demonstrated that a significant amount of PVDF binder was retained, with the characteristic peaks representing PVDF also showing a strong peak on the CSR-treated samples. We reinterpreted the TG-DTG data in more detail in line 206-218 in the revised manuscript, and the specific and corresponding Supplementary Fig. 8 and Fig. 2c are attached below:

“In addition, the decomposition of by-products and organic components in CEI layer are also supported by the thermogravimetric (TG) analysis. The TG and DTG profiles of the electrodes are displayed in Supplementary Fig. 8. All samples had two wide DTG peak groups at 350 and 510 °C, representing two different stages of weight loss. The first weight loss stage from 300 to 420 °C can be assigned to a complex residual carbonate on the surface [Journal of Power Sources (1998) 74(2) 240-243] at 330°C and part of PVDF binder decomposition occurs at 350°C [Chemical Engineering Journal, 2020, 383: 123089.; ACS Sustainable Chemistry & Engineering, (2018), 10896-10904, 6(8)]. The second weight loss stage from 460 to 560°C belongs to the decomposition of conductive carbon black (Super-P) [Journal of cleaner production, 2015, 108: 301-311.]. For the LMRO-50C, a weight loss peak appeared early at 340°C, which can be assigned to the decomposition of partial CEI components. Although the working temperature of the CSR process already reaches close to the decomposing temperature of PVDF, the decomposition of the PVDF binder is limited due to the rapid treatment. This is also confirmed by the XPS results, where the LMRO-50CS sample still shows strong representative peak of PVDF at 688 eV in F1s spectra (Fig. 2c).”

Supplementary Fig. 8 Thermogravimetric and derivative of the thermogravimetric (TG-DTG) curves of LMRO (a), LMRO-S (LMRO after CSR treatment) (b), LMRO-50C (c), LMRO-50CS(d). Temperature rise rate of 10 degrees per minute.

Fig. 2c XPS spectra of F 1s for LMRO, LMRO-50C and LMRO-50CS

The particle for TOF-SIMS analysis shown in Supplementary Fig. 7 is random selected, so it is hard to come to stand for a statistical result, where CSR induces cracks and leading to severe deterioration at high voltages. The ex-situ XRD characterization (as is shown in Fig. 3e and Supplementary Fig. 11) demonstrate that the variation on crystal

lattice is limited for the LMRO-50CS sample, so the occurrence of crack formation may be limited. Thus, the cycling performance is more stable for LMRO-50CS sample.

Minor rectifications:

(1) For clarity, legends for Fig. 4 and 5 should be made clearer. Annotate corresponding voltages with charge or discharge states, for instance: 50CS_3.2V → 50CS_3.2V (Discharge), 50CS_4.8V → 50CS_4.8V (Charge)

Answer: Thanks for the suggestions from the reviewer. In order to more clearly represent the voltages corresponding to different samples and their charging and discharging states, we have modified the representation in the legend of Fig. 4a-d and Fig 5c-e in the revised manuscript. Where the (CC) represent charging state and (DC) represent discharging state.

(2) Please add the notation "O K-edge" to the EELS data in Fig. 5a-b.

Answer: Thanks for the suggestion from the reviewer. We have added the notation "O K-edge" in Fig. 5a-b in the revised manuscript.

(3) Correct the misspellings: use "CSR" instead of "FSI" and "LMRO" rather than "LRMO" in Supplementary Fig. 3.

Answer: Thanks for the corrections. We have corrected the above spelling errors and updated them in the revised manuscript.

REVIEWERS' COMMENTS

Reviewer #1 (Remarks to the Author):

The authors have addressed my concerns. I support the publication of this work.

Reviewer #2 (Remarks to the Author):

The authors have addressed all of my concerns. The methodology used to analyze their data is now much clearer. I believe the manuscript should be accepted for publication.